Distributions of extinction times from fossil ages and tree topologies: the example of mid-Permian synapsid extinctions

http://orcid.org/0000-0003-0596-9112 Didier Gilles 1 gilles.didier@umontpellier.fr
http://orcid.org/0000-0003-2974-9835 Laurin Michel 2
1 Univ Montpellier, CNRS, IMAG , Montpellier , France
2 CNRS/MNHN/UPMC, Sorbonne Université, Muséum National d’Histoire Naturelle, CR2P (“Centre de Recherches sur la Paléobiodiversité et les Paléoenvironnements” UMR 7207) , Paris , France
Provete Diogo
Electronic publication date: 2021 Dec 9
Publication date: 2021
Volume: 9
Electronic Location ID: e12577
Received 2021 Jun 14; Accepted 2021 Nov 9
Copyright: © 2021 Didier and Laurin
Copyright year: 2021
Copyright holder: Didier and Laurin
License: This is an open access article distributed under the terms of the Creative Commons Attribution License, which permits unrestricted use, distribution, reproduction and adaptation in any medium and for any purpose provided that it is properly attributed. For attribution, the original author(s), title, publication source (PeerJ) and either DOI or URL of the article must be cited.
License URL: https://creativecommons.org/licenses/by/4.0/

Keywords: Extinction time, Fossilized-birth-death model, Permo-Carboniferous synapsids, Mass extinction events, Permian, Ophiacodontidae, Edaphosauridae, Sphenacodontidae, Amniotes, Fossil record

Funding: The authors received no funding for this work.

==============================
Given a phylogenetic tree that includes only extinct, or a mix of extinct and extant taxa, where at least some fossil data are available, we present a method to compute the distribution of the extinction time of a given set of taxa under the Fossilized-Birth-Death model. Our approach differs from the previous ones in that it takes into account (i) the possibility that the taxa or the clade considered may diversify before going extinct and (ii) the whole phylogenetic tree to estimate extinction times, whilst previous methods do not consider the diversification process and deal with each branch independently. Because of this, our method can estimate extinction times of lineages represented by a single fossil, provided that they belong to a clade that includes other fossil occurrences. We assess and compare our new approach with a standard previous one using simulated data. Results show that our method provides more accurate confidence intervals. This new approach is applied to the study of the extinction time of three Permo-Carboniferous synapsid taxa (Ophiacodontidae, Edaphosauridae, and Sphenacodontidae) that are thought to have disappeared toward the end of the Cisuralian (early Permian), or possibly shortly thereafter. The timing of extinctions of these three taxa and of their component lineages supports the idea that the biological crisis in the late Kungurian/early Roadian consisted of a progressive decline in biodiversity throughout the Kungurian.

Introduction

Reconstructing the history of the diversification of life on Earth has long been one of the main goals of evolutionary biology. In this great enterprise, the fossil record plays a central role because it gives direct evidence (even if fragmentary) of the biodiversity at various times (Carroll, 1988). It even documents spectacular changes in the rates of cladogenesis (evolutionary divergence of a lineage that splits into two lineages, a process that we here equate with speciation), anagenesis, and extinction, which occurred more or less frequently in the history of life. These were often caused by environmental changes, some of which may have resulted from intense volcanism (Wignall et al., 2009), impacts of large meteorites (Basu et al., 2003; Tabor et al., 2020), a combination of both (Arens & West, 2008), or simply transgressions, regressions (Hallam, 1989), or glaciations, among others. Such changes are associated with evolutionary radiations (Ronquist et al., 2012; Slater, 2015; Gavryushkina et al., 2016; Brocklehurst, 2017; Ascarrunz et al., 2019) that occur when the diversification of a taxon accelerates significantly, and mass extinction events (Axelrod & Bailey, 1968; Lewin, 1983; Raup & Sepkoski, 1984; Stanley, 1988; MacLeod, 1996; Benton, 2003; Ward et al., 2005; Retallack et al., 2006; Wignall et al., 2009; Bond et al., 2010; Ruta et al., 2011; Sidor et al., 2013; Lucas, 2017; Brocklehurst, 2018), during which the extinction rate of many taxa increases greatly, but typically for a short time.

So far, most studies of these phenomena that emphasized the fossil record have used the taxic approach, which consists of counting the number of taxa of a given rank (most frequently, families or genera; more rarely, species) in various time bins and estimating fluctuations in extinction and origination rates (Raup & Sepkoski, 1984; Benton, 1985; Benton, 1989; Alroy, 1996; Day et al., 2015b; Brocklehurst, 2018). Several limitations are inherent to this approach.

First, some early studies relied on databases that included many paraphyletic or even polyphyletic taxa, and thus confused pseudoextinction with genuine extinction (Patterson & Smith, 1987), even though the extinction of a paraphyletic taxon often coincides with the extinction of several smaller clades therein. Indeed, this was identified by Benton (1989) as one of the main aspects that could be improved in subsequent studies, and subsequent developments proved him right (Uhen, 1996; Fara, 2004; Marjanović & Laurin, 2008). Some recent analyses using the taxic approach even include a phylogenetic correction to these biodiversity counts by accounting for ghost lineages (Ruta et al., 2011; Martin et al., 2014; Jouve et al., 2017). Progress has also been made on how to integrate events from various sections into an optimal sequence (Sadler, 2004), and this has been applied to one of the Permian mass extinction events (Day et al., 2015b).

Second, counts of taxa at a given Linnaean nomenclatural level (except for species, if these are conceptualized as an evolutionary lineage) are poor measures of biodiversity (Bertrand, Pleijel & Rouse, 2006; Laurin, 2010), because taxa of a given level (i.e., Linnaean rank) share no objective properties (Ereshefsky, 2002) and may include one to many lineages. For this reason, better ways to quantify biodiversity were developed (Faith, 1992). We do not imply that lineage-level analyses provide a complete picture of evolution of biodiversity because other aspects are relevant, such as disparity. Some key evolutionary events, such as the Cambrian explosion, may be remarkable because of the increase in disparity rather than in speciation rate (which is poorly constrained in the Ediacarian). To an extent, counts of taxa of higher ranks may capture this, but in a indirect and imprecise way, and the evolution of disparity is better tackled by quantitative measures designed specifically to capture this (see, e.g., Wilson et al., 2013).

A third problem of the classical taxic approach is that the known stratigraphic ranges of taxa typically underestimate their true stratigraphic range (real age of appearance and extinction), a problem that is likely to be especially acute for taxa with a poor fossil record (Strauss & Sadler, 1989). Most recent analyses using the taxic approach attempt to compensate indirectly for the incompleteness of the fossil record (see, e.g., Foote, 2003; Lu, Yogo & Marshall, 2006), but this does not yield a clear idea about the timing of exinction of individual lineages.

Fourth, counting taxa in time bins can create two types of artefacts. First, if the time bins are relatively long (like geological periods or stages), the resulting counts may give the impression that origination or extinction events are concentrated at the limits between two consecutive bins, whereas in fact, the diversity of a taxon may have changed more or less gradually throughout the time bin (Day et al., 2015b; Lucas, 2017). Some methods have been devised to minimize this problem (Foote, 2003), but when more detailed stratigraphic data are available, other methods may be more appropriate to better assess whether the changes are abrupt or gradual. However, this raises another problem: for taxa that have a sparse fossil record, simple sampling effects may give the false impression that extinctions have been gradual (which is the second type of artefact evoked above). This is called the “Signor-Lipps effect” because of the landmark study by Signor & Lipps (1982), even though Shaw (1964) described it earlier, according to MacLeod (1996). Again, some taxic studies have tackled this problem to an extent (Lu, Yogo & Marshall, 2006), but these methods yield data on taxonomic global turnover rates in a given time interval, rather than a fine-scale view of the timing of extinction of individual taxa.

To establish a better understanding of the dynamics of fluctuations in biodiversity over time, it is thus useful to assess as accurately as possible the stratigraphic ranges of taxa. Early developments in this field tackled both ends (origination and extinction) of the stratigraphic ranges of taxa (Strauss & Sadler, 1989; Marshall, 1990; Marshall, 1997; Wagner, 2000). Most recent methodological developments have addressed the problem of taxon origination by inferring how much of the earliest phase of each taxon’s history remains hidden from the known fossil record, which may be useful to date the nodes of the Tree of Life (Tavaré et al., 2002; Marshall, 2008; Laurin, 2012; Warnock, Yang & Donoghue, 2012; Sterli, Pol & Laurin, 2013; Warnock et al., 2015; Didier & Laurin, 2020). However, determining when taxa became extinct is also interesting, especially to better understand past biological crises. Mass extinction events have been increasingly studied in the last decades, especially for the end-Permian event (e.g., Benton, 2003; Ward et al., 2005; Retallack et al., 2006; Wignall et al., 2009; Bond et al., 2010; Ruta et al., 2011; Sidor et al., 2013; Lucas, 2017; Brocklehurst, 2018), a trend that is partly fueled by the rising concern about the current anthropogenic biodiversity crisis (Wake & Vredenburg, 2008; Barnosky et al., 2011; Ceballos et al., 2015; Ceballos, Ehrlich & Dirzo, 2017). Thus, time is ripe to return to the question of timing of extinction of taxa.

Most of the approaches that addressed this question were derived from the seminal work of Strauss & Sadler (1989), which can provide confidence intervals for the origination and extinction time of a taxon when its fossilization potential is constant in time. Later, Marshall (1994, 1997) and Marjanović & Laurin (2008) extended this work to the case where the fossilization potential varies through time. In the same way, Silvestro et al. (2014; Silvestro, Salamin & Schnitzler, 2014) considered a model where the fossilization recovery rate follows a PERT distribution (a generalized form of the beta distribution) between the origin and the end of a lineage, which is used in a Bayesian framework with priors defined from a birth-death process, to estimate the speciation and the extinction times. Wang et al. (2016) developed a method to estimate the recovery potential function from the fossil occurrences without a priori assumptions. Among the approaches which are not derived from that of Strauss & Sadler (1989), let us mention that of Bradshaw et al. (2012), which is based on the method of McInerny et al. (2006) and that of Alroy (2014). We refer to Laurin (2012), Wang & Marshall (2016) and Marshall (2019) for recent reviews on this topic.

Studies of mass extinction events focus on patterns affecting taxa of differing richnesses, some of which include many lineages. For such studies, the methodology of Strauss & Sadler (1989) and of previous approaches might not be appropriate because they consider each taxon as if it were composed of a single lineage that does not diversify. Thus, the derivative approaches do not take into account the possibility that the considered taxon (whether it is composed of one or more lineages) may have given birth to one or more lineages that left no fossil record before going extinct, possibly well after all lineages documented in the fossil record (Fig. 1). Neglecting this possibility could be justified in the case where the diversification rates are low with regard to the fossil recovery rate. Unfortunately, our previous studies suggest the opposite situation in the datasets considered in Didier, Fau & Laurin (2017) and Didier & Laurin (2020). It follows that one could expect the extinction times (and the stratigraphic range extensions) provided by Strauss & Sadler (1989) to be inaccurate in some cases. This problem may be minor when estimating the stratigraphic range of a single nominal species, but it is probably more severe when estimating the extinction time of a clade known to have included several species, as in the case of the Permo-Carboniferous taxa (Ophiacodontidae, Edaphosauridae and Sphenacodontidae) studied below. Note that taking into account the diversification process to assess the time of extinction is much more important for a clade that became extinct long ago (i.e., tens of thousands of years ago or more) than for lineages that became extinct in historical times and for which sighting records are available (Rivadeneira, Hunt & Roy, 2009). The timescales involved in the latter case ensure that no speciation (cladogenesis) event may occur between the last observation and the extinction of the considered lineage.

Figure 1 A simulated extinct clade with sampled fossils represented by brown dots.

Top: The clade’s complete evolutionary history. Bottom: The portion of the clade’s history observable from the known fossil record. Note that the ‘blue’ and ‘yellow’ taxa diversify before going extinct, but that these diversification events are not recorded in the known fossil record.

To better estimate extinction time by considering additional lineages that may have left no fossil record, the fossilized birth-death model (FBD model) could be used. The FBD model assumes that fossil finds follow a Poisson process, which is also assumed by Strauss & Sadler (1989), but it also models the diversification of taxa as a birth-death process. Given that the parameters that characterize the FBD model include an extinction rate, it should be possible to use this process to estimate the probability distribution of extinction times. So far, the FBD model has been used to date cladogenetic events (Stadler & Yang, 2013; Heath, Huelsenbeck & Stadler, 2014; Didier & Laurin, 2020) but usually not extinction, with the exception of Brocklehurst (2020), who used tip dating with the FBD to assess Olson’s extinction and to reject the idea of Olson’s gap. Evaluating extinction times through the FBD would be very useful to determine the extent to which the Signor-Lipps effect has biased our perspective on mass extinction events. It could also be useful to reassess the reliability and stratigraphic significance of some taxa as index fossils, at least those with a relatively sparse fossil record for which reliable phylogenies exist; such cases are presumably fairly rare in the marine realm, but may be more common in continental biochronology (Steyer, 2000; Day et al., 2013, 2015a). Indeed, stratigraphic correlations of continental strata, at least when relying on vertebrate fossils, often use higher-ranking taxa (nominal genera or families), especially when strata located on different continents are assessed (Rubidge, 2005; Lucas, 2018; Lucas & Shen, 2018).

Another concern with previous approaches is that they require several fossils of a taxon to provide a confidence interval that bounds the corresponding extinction time, which makes it unsuitable for lineages or small clades with a low fossilization rate. Moreover, because it is computed independently on each extinct taxon (without consideration of its close relatives and the tree structure of this set of lineages), the level of precision provided by the previous methods depends on the number of fossils present on each terminal branch. This point can be a major issue for datasets where the fossil recovery is low (see Simulation Study below). This limitation does not apply to the method we propose, in which the extinction-time distribution of taxa with a single fossil on their terminal branch can be determined, if this branch belongs to a clade with a sufficient number of fossils. This results largely from the fact that all the data in a given dataset (which must represent a clade that may be truncated at any given time in the past) are used to assess the FBD parameters and hence, are considered in the computations of extinction time densities of all its branches.

Below, we extend the FBD model to estimate extinction times of taxa that may consist of one to many lineages. Specifically, given a dataset that consists of a phylogenetic tree of just extinct, or extinct and extant taxa, where at least some fossil data are available but without divergence times, we compute the probability that a given set of taxa (known to be extinct at the present time) goes extinct before a time t. The computation of this probability density is a direct extension of the method provided in Didier & Laurin (2020). We also provide an explicit formula for the probability that a given subset of extinct taxa (typically a clade) goes extinct before another one under the FBD model (Section S2). We adapted the Monte Carlo Markov Chain (MCMC) importance sampling procedure devised by Didier & Laurin (2020) in order to deal with the common case where fossil ages are provided as stratigraphic time intervals and to integrate distributions over the parameters of the model. Our approach allows incorporating phylogenetic uncertainty by estimating the extinction times over a set of trees.

The method presented below, like our previous works in this field, assumes a homogeneous fossilization rate through time. This assumption about the quality of the fossil record is made by most methods based on the FBD model. In practice, this requirement is never met, but mild violations of this assumption, like random fluctuations through time, should affect the reliability of the method less than pervasive trends over the considered time interval. Note that this hypothesis would not be reasonable if we had adopted what Lucas (2017) called the “best sections” analysis, which focuses on a region where a given taxon has a rich fossil record for a given period. Given that the FBD models diversification of clades, and that the clades that we analyze had a cosmopolitan distribution, it would be inappropriate to restrict our analysis to a single fossiliferous sedimentary basin. Note also that this assumption is analogous with the molecular clock, which initially was assumed, for computation purposes, to be strict and universal (Zuckerkandl & Pauling, 1965), before local or relaxed clock methods (Cooper & Penny, 1997; Sanderson, 2002; Drummond et al., 2006) were developed to account for rate variations that had been suspected to occur from the very beginning. Likewise, the FBD is probably amenable to such developments, but these are beyond the scope of this study.

Our approach is first assessed and compared to that of Strauss & Sadler (1989) and three other approaches on simulated datasets. It is then applied to study the extinction of three Permo-Carboniferous taxa: Ophiacodontidae, Edaphosauridae and Sphenacodontidae.

The computation of the extinction time distribution and of its confidence upper bound at a given threshold was implemented as a computer program and as a R package, both available at https://github.com/gilles-didier/DateFBD and in the Supplemental Material.

Empirical example: Permian extinction of ophiacodontidae, edaphosauridae and sphenacodontidae

We illustrate our method with an empirical example from the rich fossil record of Permian synapsids. Synapsida originated in the Carboniferous and experienced a few evolutionary radiations, the first one of which, in the Late Carboniferous and early Permian, gave rise to taxa that have long been known as “pelycosaurs” (Romer & Price, 1940; Reisz, 1986; Benson, 2012), but which will be called here “Permo-Carboniferous synapsids”. Among these taxa arose the stem lineage of therapsids, probably in the Late Carboniferous (Sidor, 2001; Amson & Laurin, 2011; Spindler, 2014; Angielczyk & Kammerer, 2018). Therapsids become increasingly common in the Roadian (early middle Permian) fossil record (Reisz & Laurin, 2002; Abdala, Rubidge & Van Den Heever, 2008), and dominated several ecological niches from the Wordian (mid-Guadalupian) to the end of the Permian (Smith, Rubidge & Van der Walt, 2012). All other synapsid clades appear to have became extinct before the end of the Guadalupian (Modesto et al., 2011). Therapsida experienced several evolutionary radiations, including one that gave rise to mammals, in the Triassic or in the Jurassic (King & Beck, 2020).

Up to four mass extinction events have been recognized in the Permian fossil record of synapsids (Lucas, 2017), and a brief review of these is relevant to understand the context of the present study and to justify the taxonomic sample. The first may have occurred at the Artinskian/Kungurian boundary (about 282 Ma), or possibly at the Sakmarian/Artinskian boundary (about 290.1 Ma), or it may be a long decline that occurred throughout the Sakmarian and Artinskian (Benton, 1985, 1989; Brocklehurst, Kammerer & Fröbisch, 2013). This extinction was suggested by early studies. Olson & Vaughn (1970), and Brocklehurst, Kammerer & Fröbisch (2013) mentioned it too. Benton (1989) stated that Ophiadocontidae, Edaphosauridae, and Sphenacodontidae were among the taxa that became extinct then. However, Lucas (2017) argued that it represents a normal level of faunal turnover. In any case, it is now clear that Ophiadocontidae, Edaphosauridae, and Sphenacodontidae persisted at least until the Kungurian.

The second possible mass extinction event in Permian tetrapods, which we study here, may have occurred near the Kungurian/Roadian stage boundary (Lucas, 2017; Brocklehurst, 2018), which is also the Cisuralian (early Permian)/Guadalupian (middle Permian) series boundary (272.3 Ma). Some of the extinctions (among others, those of Ophiadocontidae, Edaphosauridae, and Sphenacodontidae) that had at some point been postulated to have taken place in the Sakmarian and/or Artinskian may have taken place toward the end of the Kungurian, or slightly later. The observed stratigraphic range of Ophiadocontidae and Edaphosauridae ends shortly before the top of the Kungurian, whereas Sphenacodontidae may well extend into the early Roadian, given the controversial and poorly constrained age of the San Angelo Formation (see below). Sahney & Benton (2008) called this event “Olson’s extinction”. They did not date it very precisely, mentioning only that it had taken place in the Roadian (272.3–268.8 Ma) and/or Wordian (268.8–265.1 Ma), but estimated that it “reveals an extended period of low diversity when worldwide two-thirds of terrestrial vertebrate life was lost” (Sahney & Benton, 2008, p. 760). Olroyd & Sidor (2017) suggested, among other hypotheses, that extinction of most Permo-Carboniferous synapsids at the end of the Kungurian could have allowed their non-competitive replacement by therapsids, which is supported by the results of Sahney & Benton (2008) and Brocklehurst, Kammerer & Fröbisch (2013). This hypothesis is tested indirectly here (see below). Lucas (2018, p. 430) suggested that rather than a single large crisis, a few events (which he called “Redtankian events”, after his Redtankian chronofauna) took place in the Kungurian. Lucas (2018, p. 430) suggested that ophiacodontids became extinct before edaphosaurids, but that both clades were extinct in the early Redtankian. Sphenacodontids became extinct later because they occur in the Littlecrotonian Lucas (2017, p. 43). However, this is based on a literal interpretation of the fossil record; no attempts have been made at assessing confidence intervals for the extinction times of relevant taxa, as far as we know, although Brocklehurst, 2018; Brocklehurst (2020) studied “Olson’s extinction” through other methods. Our study aims at filling this gap.

The third mass extinction event took place near the end of the Capitanian (259.8 Ma), the last stage of the Guadalupian (middle Permian), around the time of the Emeishan volcanism in southern China (Day et al., 2015b; Lucas, 2017). It wiped out the dinocephalians (a fairly large clade of Guadalupian therapsids), although this apparently occurred gradually in the Tapinocephalus Assemblage Zone (Day et al., 2015a). Other therapsid taxa also appear to have been affected by this crisis, which apparently also influenced the marine realm (Day et al., 2015b). Varanopid synapsids also appear to have become extinct then (Modesto et al., 2011); this was the last of the Permo-Carboniferous synapsid clades to become extinct because caseids are not currently known after the Roadian. Maddin, Sidor & Reisz (2008) assigned a mid-Capitanian age to the caseid Ennatosaurus tecton, but more recently, Golubev (2015) assigned a Roadian age to the locality (Moroznitsa) of the holotype. Lucas (2017) argued that parareptiles were also affected by this crisis, but other studies suggests that parareptiles were only affected by background extinctions throughout the Guadalupian (Ruta et al., 2011; Cisneros et al., 2020). Day et al. (2015b) estimated that there was a 74–80% loss of generic richness in amniotes in this crisis, and that it was not as severe as the end-Permian crisis. On the contrary, Lucas (2017) considered that this extinction event was more severe for amniotes than the much better-known end-Permian event and that both may have lasted longer than previously thought.

The fourth and best-known of the Permian mass extinction event took place at the Permian/Triassic boundary (251.9 Ma). It was once thought that this crisis lasted over the last 10 Ma of the Permian (Erwin, 1990), but that was before the end-Guadalupian crisis was identified. More recent studies point to a much shorter crisis. It has been estimated that between 80% and 96% of the marine species were eliminated (Sahney & Benton, 2008), but recent studies have shown a significant crisis in continental vertebrates as well (Ward et al., 2005). Thus, Smith, Rubidge & Van der Walt (2012, p. 47) reported that out of 41 therapsid genera present in the middle of the Dicynodon Assemblage Zone, only three survived the end-Permian extinction (excluding ghost lineages that imply a greater proportion of survivors). Lucas (2017) claims that this extinction event lasted longer than previously claimed and occurred in a stepwise manner. Among recent works on this topic, Viglietti et al. (2021) differs by using a fine stratigraphic resolution of 13 time bins lasting about 300,000 years each, thus covering in detail evolution of tetrapod biodiversity in the Karoo around the P/Tr boundary. This work showed the advantages of using a fine stratigraphic scale to better understand mass extinction events. This showed that the crisis among continental tetrapods of the Karoo Basin lasted about 1 Ma, which is longer than the marine crisis, which is believed to have lasted only about 61,000 years (Burgess, Bowring & Shen, 2014; Liu et al., 2020).

As can be seen from this very brief review, a recurring question is how long each crisis lasted. In other words, did most taxa become extinct at about the same time (in a few tens of thousand years, possibly a few hundred thousand years), or were the extinctions spread over a few million years? Settling this question ideally requires abundant, well-dated, and geographically-widespread data, as well as appropriate analytical methods to discriminate between genuine gradual extinctions and the Signor-Lipps effect on taxa with a relatively scarce fossil record, as is typically the case for continental vertebrates.

Our dataset is relevant to assess extinction times that span the first and second of the possible tetrapod Permian mass extinction events. Our method is not designed to assess fluctuations in extinction rates; rather, our objective is to obtain a better understanding of the timing of extinction of various taxa to either corroborate or refute previous statements about such events. More specifically, we test the following hypotheses: Many ophiacodontids, edaphosaurids, and sphenacodontids had become extinct well before the end of the Kungurian (which is consistent with a prolonged crisis, or a series of crises, rather than with a single, catastrophic, sudden event at the end of the Kungurian; this can also test the existence of the first of the four crises listed above, near the Artinskian/Kungurian boundary).

Ophiadocontidae, Edaphosauridae, and Sphenacodontidae became extinct (gradually or not) by the end of the Kungurian, at the latest (Benton, 1989; Brocklehurst, Kammerer & Fröbisch, 2013; Lucas, 2017);

These three clades became extinct in the following order: Ophiacodontidae, Edaphosauridae, and Sphenacodontidae (Lucas, 2018, p. 430).

We test the first hypothesis by verifying the proportion of terminal branches (observed nominal species) of these three clades that became extinct before the end of the Kungurian (i.e., more than 95% of their extinction probability is before the end of the Kungurian, which is 272.3 Ma). A substantial proportion of lineages becoming extinct before the end of the Kungurian would be compatible with a gradual extinction of these clades, even though additional tests will be required to prove this hypothesis. If a high proportion of the early extinctions were concentrated in time (especially around the Artinskian/Kungurian boundary), this would be compatible with (but would not prove) the possibility of a crisis around that time.

We test the second hypothesis by verifying if the extinction-density probability of the three clades is compatible with an extinction of these clades by the end of the Kungurian. Contrary to previous methods, ours considers the extinction times of lineages that have not been preserved in the fossil record but that are very likely to have existed because of the speciation, extinction and fossilization rates. The fact that Didier, Fau & Laurin (2017, p. 981) estimated that only about 14% of the eupelycosaur lineages (defined as an internode on the tree) had left a fossil record suggests that taking into consideration unobserved lineages can have a major impact on our estimates of the extinction times of these three large eupelycosaur clades. Hence, these times should be somewhat later than the extinction of the most recent known lineage of each of these clades, and potentially, substantially more recent than the last observed fossil of each of these clades.

The third hypothesis is tested by looking at the peak probability density of extinction time and the end of the 95% confidence interval of the extinction time of all three clades and by computing the probability of a given clade becoming extinct before another clade under the FBD model.

Note that testing hypotheses 1 and 2 above amounts to testing indirectly the suggestion that the replacement of Permo-Carboniferous synapsids by therapsids was non-competitive (Olroyd & Sidor, 2017, p. 593) because a literal reading of the fossil record suggests that therapsid diversification accelerated sharply in the Roadian and that this is after the extinction of Ophiacodontidae, Edaphosauridae, and Sphenacodontidae. If this is correct, this replacement was non-competitive. On the contrary, if Ophiacodontidae, Edaphosauridae, and Sphenacodontidae became extinct only in the Wordian or later (time at which theraspids were already abundant in most terrestrial assemblages), this would suggest a competitive replacement.

These three hypotheses are not mutually exclusive. For instance, these clades could have become extinct gradually by the end of the Kungurian in the order suggested by Lucas (2018, p. 430), in which case all three hypotheses would be correct. Alternatively, these three clades might also have become extinct suddenly after the Kungurian and in a different sequence than stipulated by Lucas (2018, p. 430), even though this is not what a literal interpretation of their fossil record suggests; in this case, all three hypotheses would be false. But any combination is possible.

Corrections for multiple tests are not required in this study because the goal of the paper is not to test n times that n species became extinct before the Kungurian/Roadian boundary (which would indeed amount to making a high number of tests). Rather, we want to get a picture of the pattern of extinction (gradual vs simultaneous) and assess where it fits compared to the Kungurian/Roadian boundary. The actual number of tests (for which we report probabilities) performed in this study is fairly low.

Methods

Methods to estimates extinction times which are presented below all return a confidence upper bound of the extinction date at a given order (95% is the usual choice), i.e., the time t which is such that the probability for the extinction date to be anterior to t is equal to the order required.

Previous approaches

Most of previous approaches estimate extinction times “branch by branch” in the sense that they deal with each taxon independently, by taking into account only its own fossils to estimate its extinction time. Among those, the approach of Strauss & Sadler (1989) assumes that the fossil ages are uniformly distributed during the lifetime of a taxon to provide a confidence upper bound of its extinction time. The approach of McInerny et al. (2006) makes a similar assumption of uniformity but considers discretized time to infer extinction ages from sighting records. This last approach can be applied to the fossil case as well and is the basis of the method proposed by Bradshaw et al. (2012), which modified the approach of McInerny et al. (2006) by giving more weights to the most recent fossil ages and by taking into account the uncertainty in the fossil ages. The approach of Alroy (2014) also discretizes time into equal subintervals to infer extinction ages with a Bayesian iterative approach. In this study, we shall consider a continuous version of the approach of McInerny et al. (2006), from which we have derived a continuous global version of this method. This last approach is global in the sense that it takes into account all the fossil ages of the phylogenetic tree, not only those of the considered taxon, to infer extinction times. The presentation of the approaches above is further detailed in the Supplemental Information (Section S1).

The FBD model

The FBD model was introduced in Stadler (2010) and has been referred to as the “Fossilized-Birth-Death” model since Heath, Huelsenbeck & Stadler (2014). This model assumes that the diversification process starts with a single lineage at the origin time, which is one of its parameters. Next, each lineage alive evolves independently until its extinction and may be subject during its lifetime to events of speciation (here equated with cladogenesis, which leads to the birth of a new lineage), extinction (which terminates the lineage) or fossilization (which leaves a fossil of the lineage dated at the time of the event) which occur with respective rates λ, μ and ψ, which are the main parameters of the model. Last, the extant lineages (if any) are sampled at the present time with probability ρ, the last parameter of the model.

Let us recall the probabilities of the following basic events, derived in Stadler (2010), Didier, Royer-Carenzi & Laurin (2012) and Didier, Fau & Laurin (2017) under the FBD model, which will be used to compute various probability distributions on the observable part of realizations of the FBD process (Fig. 1 here or Didier & Laurin (2020: Fig. 1). The probability P(n,t) that a single lineage starting at time 0 has n descendants sampled with probability ρ at time t > 0 without leaving any fossil (i.e., neither from itself nor from any of its descendants) dated between 0 and t is given by

P(0,t)=α(β−(1−ρ))−β(α−(1−ρ))eωtβ−(1−ρ)−(α−(1−ρ))eωtand

P(n,t)=ρn(β−α)2eωt(1−eωt)n−1(β−(1−ρ)−(α−(1−ρ))eωt)n+1foralln>0,

where α < β are the roots of − λ x2 + (λ + μ + ψ)x − μ = 0, which are always real (if λ is positive) and are equal to

λ+μ+ψ±(λ+μ+ψ)2−4λμ2λ,

and where ω = − λ(β − α). Note that P(0,t) is the probability that a lineage alive at time 0 goes extinct before t without leaving any fossil dated between 0 and t.

The probability density D(t) for a lineage alive at time 0 to go extinct exactly at time t (without leaving any fossil) is basically obtained by deriving P(0,t) with regard to t. We get that

D(t)=λ(α−(1−ρ))(β−(1−ρ))(β−α)2eωt(β−(1−ρ)−(α−(1−ρ))eωt)2.

In the particular case where ρ = 1, expressions above simplify to

(1) P(0,t)=αβ(1−eωt)β−αeωt ,P(n,t)=(β−α)2eωt(1−eωt)n−1(β−αeωt)n+1foralln>0 and D(t)=μ(β−α)2eωt(β−αeωt)2.

Extinction time distributions

We consider here a dataset consisting of a phylogenetic tree T of extant and extinct taxa with fossils, which is interpreted as the observable part of a realization of the FBD process, and of the fossil age vector f. We aim to compute the joint probability density of (T,f) and that a particular subset S of taxa of T goes extinct before a time t under the FBD model. This question makes sense only if the taxa of S all go extinct before the present time. In the empirical example provided below, these subsets consist of three clades, but the method can handle any set (monophyletic or not) of taxa.

This joint probability density can be computed in the exact same way as that of the dataset (T,f) presented in Didier & Laurin (2020). To show this, we adapted figure 3 from Didier & Laurin (2020), which illustrates the fact that, from the Markov property, the probability density of (T,f) can be written as the product of that of the “basic trees” obtained by splitting T at each fossil find (Fig. 2). The tree of Fig. 2 contains only one extinct taxon that goes extinct between f1, the age of its most recent fossil, and the present time T. The contribution factor of the basic tree starting from time f1, which is unobservable from f1 to the present time, is the probability that a lineage alive at f1 goes extinct before the present time T without leaving any fossil dated between f1 and T, i.e., P(0, T − f1). Computing the joint probability density of the dataset of Fig. 2 and that its extinct lineage goes extinct before a given time t (resp. exactly at a given time t) is performed by setting the contribution factor of the basic tree starting from time f1 to P(0, t − f1) (resp. to D(t − f1)).

Figure 2 Decomposition of the probability density of a phylogenetic tree of extinct and extant taxa with fossils (figured by brown dots) as the product of the probability densities of “basic trees” by cutting it at each fossil find (adapted from Didier & Laurin, 2020).

Let us now consider a subset S of extinct taxa of T, for instance, an extinct clade of T. The joint probability density of the dataset (T,f) and that S goes extinct before a given time t, which is basically the joint probability density of (T,f) and that all the taxa of S goes extinct before t, is obtained by setting, for all taxa n ∈ S, the contribution factor of the unobservable basic tree pending from the leaf n, which starts from time lf,n, i.e., the age of the most recent fossil of n, to P(0, t − lf,n), that is the probability that a lineage alive at f,n goes extinct before t without leaving any fossil dated between lf,n and t. The computation of this joint probability density is thus computed in the very same way and with the same algorithmic complexity as that of the probability density of (T,f) presented in Didier & Laurin (2020).

The computation presented above allows us to determine the probability distribution of the extinction time of a subset of taxa in the case where the tree topology, the fossil ages and the parameters of the FDB model are exactly known, a situation which is unfortunately never met in practice. Namely, the rates of the FBD model are unknown, morphological data of the fossils may be consistent with several topologies, and fossil ages are provided as time intervals of plausible ages.

Given a priori probability distributions over the parameters and the data, a standard way to deal with the uncertainty on the model parameters and the data consists in integrating the extinction time distribution over all the possible values of the FBD parameters, of the fossil ages (constrained within confidence intervals) and over all tree topologies included in a population of trees. We implemented the numerical computation of this integration by Monte Carlo Markov Chain (MCMC, see Section S3).

Note that determining the whole distribution of the extinction time is generally not required. Getting a confidence upper bound of the extinction date at a given order (usually 95%) provides sufficient information for most purposes. In our framework, this bound is basically the quantile at the order required of the extinction time distribution.

If one is interested in comparing the extinction dates of two given extinct clades (or more generally two subsets of extinct taxa) A and B, it is possible to compute the probability that A goes extinct before B under the FBD model. In Section S2, we provide an explicit formula for this probability.

Empirical dataset

Our dataset includes a subset of the taxa represented in the dataset used by Didier & Laurin (2020), from which we simply extracted the smallest clade that includes Ophiacodontidae, Edaphosauridae and Sphenacodontidae. However, we incorporated all the information that we could find about the relevant taxa in the Paleobiology Database (Alroy, Marshall & Miller, 2012). Thus, we now have more fossil occurrences than in our previous studies that used this dataset (Didier, Fau & Laurin, 2017; Didier & Laurin, 2020). Most of the fossil occurrences in our database are from south-western North America, but a few (like Archaeothyris florensis) come from the eastern part of that continent, as well as from Europe (such as Haptodus baylei). We also updated the geological ages using the recent literature (see below). Finally, we updated our supermatrix (which is used to produce the population of source trees) to include more recent studies that update scores for various taxa and provide scores for taxa for which the earlier version of our matrix had no data (the position of these taxa was specified using a skeletal constraint). Specifically, we replaced the matrix of Benson (2012) by that of Mann & Paterson (2020), which is the most recent update of that matrix. To better resolve the phylogeny of edaphosaurids and sphenacodontids, we incorporated also the matrix from Brocklehurst & Brink (2017). This allowed us to reduce the number of taxa in the skeletal constraint, which we updated, notably to reflect current ideas about the affinities of Milosaurus (Brocklehurst & Fröbisch, 2018). The 100 trees used here were sampled randomly from three searches that each yielded 200,000 equiparsimonious trees (the maximal number of trees that we allowed in the search) that apparently include at least two tree islands, and we verified that this sample included trees of both islands. Additional searches with a lower number of maximal trees and a different seed number for the random addition sequence were carried out to verify that these were the shortest trees. We also carried out a single search with a maximal number of 15,000,000 trees. All these searches confirmed that we appear to have recovered all tree islands and that these include some of the shortest trees. All these new searches included only the taxa studied here, plus Varanops brevirostris (the varanopid with the highest number of characters scored into our supermatrix), which was used as the outgroup.

Our method, contrary to those derived from Strauss & Sadler (1989), Marshall (1994), and Marshall (1997), ideally requires stratigraphic data expressed in absolute age (Ma) rather than height in a section because we model events in time. It could use section height, provided that sedimentation rates were more or less homogeneous in time, but this would be less interesting (evolution happens in geological time, not in strata) and would be meaningless if applied to sections of different basins in which sedimentation rates were not necessarily comparable. In any case, these would still have to be correlated to each other. We thus relied on the literature to convert stratigraphic position of the fossils into approximate ages (represented by age ranges, which our method samples using a flat distribution). We tried to adopt the established consensus, as reported, for part of the fossiliferous basins in SW USA, by Nelson, Hook & Chaney (2013) and the age assignments usually attributed to these formations as interpreted (for much of the relevant stratigraphic range) by Lucas (2018, fig. 4), using the most recent geological timescale, as summarized by Lucas & Shen (2018, fig. 9) and Schneider et al. (2020). For instance, according to Lucas (2018, fig. 4), the Moran Formation extends from the latest Asselian to the late Sakmarian, and the overlying Putnam Formation extends from that time to the end of the Sakmarian (approximately). Given that Lucas & Shen (2018, fig. 9) report an age interval of 298.9 to 295 Ma for the Asselian and 295 to 290.1 Ma for the Sakmarian, we assigned age intervals of 295.5 to 291.5 Ma to the Moran Formation, and from 291.5 to 290 Ma to the Putnam Formation, and to the fossils found therein. Thus, the oldest of the two occurrences of Lupeosaurus kayi documented in the Paleobiology Database are from the Moran Formation and were thus assigned an age of 295.5 to 291.5 Ma, as can be seen in the supplements. This dating scheme admittedly involves much interpolation between strata that have been dated by radiometric methods, but it can be updated as the geological timescale is refined. As in our previous studies, we entered all documented stratigraphic horizons for all included taxa. Even if a few horizons were too close in stratigraphic height to distinguish their ages, they were entered separately, but with the same age range. For instance, Sphenacodon ferocior occurs in four localities of the Abo Formation (297 to 290.1 Ma), according to the Paleobiology Database; it has thus been scored as occurring four times in strata of this age bracket because we assume that each locality represents a distinct level. However, our analysis is carried out at the fossiliferous horizon level; even if many specimens of a given taxon were found in a given horizon, this is still scored as a single occurrence. In the absence of more detailed information, we also assume that a single level in each locality yielded each of the relevant taxa.

Contrary to Lucas (2018), we consider that most of the Pease River and El Reno Groups are Guadalupian, as previously suggested by one of us and as supported by other studies based on various types of evidence (Clapham, 1970; Reisz & Laurin, 2001; Reisz & Laurin, 2002; Nelson & Hook, 2005; Foster et al., 2014; Soreghan et al., 2015). In this study, a single relevant taxon occurs in these strata, namely Dimetrodon angelensis, in the San Angelo. We consider that the range of possible ages of this formation, which is at the base of the Pease River Group, straddles the Kungurian/Roadian boundary, from 275 Ma to 271 Ma. The stratigraphic database used here, along with the trees, is available in the supplements.

Simulation study

In order to assess the accuracy of the upper bound of the extinction time corresponding to a given confidence level (here the usual 95%) obtained from the approaches presented in Section “Methods”, we simulated the diversification and the fossilization of a clade under the FBD model with five fossilization rates. The simulated data consist of the observable parts of the simulated diversification processes but we also stored the extinction time of all the extinct taxa (including those that are not observable) in order to assess the accuracy of different approaches.

In the simulation case, we have access to the (exact) tree topology, the fossil ages and the model parameters. In this section, we shall assume that both the tree topology and the fossil ages are exactly known but consider both the confidence upper bounds obtained from the model parameters used to simulate the data, which is the reference method since the extinction ages follow the distributions from which these upper bounds are computed in this case, and the confidence upper bounds obtained without the knowledge of the model parameters, i.e., by dealing with their uncertainty by integrating over all the possible model parameters (under the assumption that they follow improper uniform distributions; Section S3).

For each simulation, the “real” extinction times of all the extinct taxa were stored and compared to the upper bounds obtained from each method. Namely, the accuracy of each method is assessed with regard to two features: their percentage of errors, that is the proportion of situations in which the real extinction date is posterior to the upper bound provided and,

the average width of the confidence interval, that is the average duration separating the age of the most recent simulated fossil of each taxon from the 95% confidence upper bound of its estimated extinction time.

The simulations were obtained by running a FDB process with speciation and extinction rates λ = 0.2 and μ = 0.19 per lineage and per million year during 200 millions years. These settings yield an expected lineage duration of about 5 millions years. Note that we simulate diversification and fossilization through evolutionary time, not through strata; thus, all simulations aim at assessing the performance of methods at inferring confidence intervals of extinction times, rather than levels in a section. We considered five fossilization rates: ψ = 0.005, 0.01, 0.1, 1 and 5. The simulations were filtered for technical reasons. We kept only those leading to an observable tree of size from 50 to 100 nodes and containing more than 20 fossils.

The six following ways to obtain a 95% confidence upper bound of the extinction date were assessed:

S&S: the method of Strauss & Sadler (1989),

Alr: the method of Alroy (2014); since several ways of computing the quantities used to determine the posterior probabilities required by this method are proposed in Alroy (2014), we tested some of them and kept those leading to the best performance (in particular, we set ps=np−1np−1+na, under the notation of Alroy (2014), which allows us to apply this approach to terminal branches with at least two fossiliferous horizons),

McI: the continuous version of the approach of McInerny et al. (2006),

Glo: the global version of McI,

Int: the 95% quantile of the extinction time distribution obtained by integrating uniformly over the speciation, extinction and fossilization rates (numerically estimated from the MCMC procedure presented in Section S3, which was applied with the following settings. For each fossil recovery rate and each of the 1,000 corresponding simulated trees, we discarded the first 10,000 iterations of the Markov chain; then, we kept 1 iteration out of 10 until getting a sample of 5,000 sets of rates from which the 95% quantiles of the extinction time distributions associated to all the extinct taxa are computed),

Ref: the 95% quantile of the extinction time distribution computed from the parameters used to simulate the data, which is the reference method (unfortunately, this method cannot be applied in practice because we do not know the actual rates of speciation, extinction and fossilization in real situations).

We did not display the results obtained by the method of Bradshaw et al. (2012) because they were systematically worse than those obtained with method McI. This was expected since the method of Bradshaw et al. (2012) is a modified version of that of McInerny et al. (2006) in a way which cannot improve performances in our simulation context, i.e., uniformity of the fossil rate and exact fossil ages.

All the methods assessed do not apply to all cases. Methods S&S, Alr and McI require at least two fossils in the branch of the considered extinct taxa while method Glo can be applied only on trees containing at least a branch bearing more than two fossils (it implies that if methods S&S, Alr and McI can be applied then so can be method Glo). Only methods Int and Ref can be applied to all extinct taxa.

Tables 1–3 display the results obtained over 1,000 simulated trees for each fossilization rate.

Table 1 Simulated extinct taxa statistics for all fossilization rates.

Column 2 displays the percentage of simulated trees which satisfy the size and number of fossils required. Columns 3 (% Branch feasible) and 4 (% Glo feasible) display the percentage of extinct taxa on which methods S&S, Alr and McI can be applied and on which method Glo can be applied, respectively. The last column shows the total number of extinct taxa observed over all the simulations. The average number of extinct taxa per tree ranges from 20 to 36 taxa.

Fos. rate	% Accepted trees	% Branch feasible	% Glo feasible	Total extinct taxa	
0.005	0.27	7.42	85.36	20,787	
0.01	0.66	10.19	95.62	25,084	
0.1	1.44	33.83	100.00	35,928	
1	2.59	75.02	100.00	36,088	
5	2.96	93.07	100.00	35,820	

Table 2 Error percentages obtained from the 95% confidence upper bounds provided by the six methods on the simulated extinct taxa where the S&S, McI and Alr computations were feasible (plain text), on those belonging to a simulated tree where a global fossil recovery rate can be estimated (italics; only for methods Glo, Ref and Int) and on all the extinct taxa (bold; only for methods Ref and Int).

Fos. rate	S&S	McI	Alr	Glo	Int	Ref	
0.005	2.40	13.35	11.34	6.48	8.02	4.93	5.30	5.38	4.67	5.07	5.12	
0.01	3.80	16.87	13.97	6.18	6.93	5.32	5.90	5.80	5.05	5.20	5.18	
0.1	3.92	18.30	11.85	4.15	4.33	5.16	5.54	5.46	4.86	4.95	4.95	
1	4.93	14.62	6.32	4.64	4.69	4.78	4.86	4.86	4.74	4.78	4.78	
5	5.27	9.40	2.09	5.17	5.18	5.20	5.20	5.20	5.11	5.10	5.10	

Table 3 Mean confidence interval width in million years, i.e., the mean difference between the 95%confidence upper bounds provided by the three methods and the most recent fossil, on the simulated extinct taxa where the S&S, McI and Alr computations were feasible (plain text), on those belonging to a simulated tree where a global fossil recovery rate can be estimated (italics; only for methods Glo, Int and Ref) and on all the extinct taxa (bold; only for methods Int and Ref).

Fos. rate	S&S	McI	Alr	Glo	Int	Ref	
0.005	244.0	40.5	36.0	40.4	41.2	26.3	25.8	25.9	26.1	26.0	26.0	
0.01	181.1	30.5	28.9	30.8	30.6	21.6	21.2	21.2	21.7	21.8	21.8	
0.1	50.5	10.3	14.0	10.3	10.3	9.0	9.0	9.0	9.2	9.2	9.2	
1	6.3	2.2	4.7	2.2	2.2	2.2	2.2	2.2	2.2	2.2	2.2	
5	0.9	0.6	1.6	0.6	0.6	0.6	0.6	0.6	0.6	0.6	0.6	

We observe that the acceptation percentages of trees are low, ranging from 0.27 to 2.96 (Table 1). One thus expects the sample of 1,000 simulated trees for each fossilization rate to be biased with regard to the corresponding FBD rates, and may worry about the effect of this bias on our analyses. Fortunately, this effect seems limited. Sections SI–S4 displays the posterior distributions of rates obtained from the MCMC and shows that they are well centered around the rate values used to simulated the corresponding trees.

As expected, Table 1 shows that the proportion of extinct taxa on which the methods S&S, McI and Alr can be applied may be very low when the fossil recovery rate is small (Table 1, Col. 3). Within the covered simulation space, this proportion ranges from about 7% when the fossil recovery rate is 4 times lower than the speciation rate, to about 93% when the fossil recovery rate is 25 times the speciation rate (note that in 3 of the 5 sets of simulations, the fossil recovery rate is much higher than the net diversification rate, which is speciation minus extinction rates). This proportion increases with the fossil recovery rate. In our simulations, the “branch-by-branch” approaches require a fossil recovery rate several times higher than the speciation and extinction rates in order to be applicable on the majority of extinct taxa. These requirements may be severely limiting because our current dataset yields fossilization rates only about twice as high as the speciation and extinction rates, and the previous version yielded much lower fossilization rates (Didier & Laurin, 2020). Even a neogene camelid dataset, which could be expected to have a much denser fossil record, only yielded fossilization rates a little more than twice as high as those of speciation and extinction (Geraads et al., 2020). There is a relatively small proportion of cases where method Glo cannot be applied, and these occur only for the two lowest fossil recovery rates (Tab. 1, Col. 4).

The total number of extinct taxa, displayed in the last column of Table 1, starts by increasing with the fossilization rate, then fluctuates around 36,000 for all the simulations when the fossilization rate is close to, or greater than the speciation and extinction rates. This saturation phenomena was expected because phylogenetic trees simulated with the speciation and extinction rates of our protocol have a certain average number of extinct branches among which only those with fossils appear as extinct taxa in the final simulated tree (Fig. 1) and the probability that an extinct branch bears at least a fossil tends to 1 as the fossil recovery rate increases.

The error percentage, that is the percentage of cases where the “real” extinction date is posterior to the confidence upper bound provided, fluctuates around 5%, which is the level of risk required here, both for methods Ref and Int (Tab. 2). This was expected for method Ref because its confidence upper bound is computed according to the actual distribution of the extinction date under the simulation model in this case. The closeness of the error percentage of method Int to the level of risk required suggests that integrating the extinction date distribution over all the possible FBD rates leads to accurate results (see also Section S4). The error percentage obtained from methods S&S, McI, Alr and Glo depend heavily on the fossil recovery rate and is generally far from the level of risk required (it is actually close to the level of risk required only for methods S&S and Glo with the two highest fossil recovery rates).

As expected, the average difference between the confidence upper bound provided by each method and the age of the most recent fossil, which we call the mean confidence interval width, tends to be reduced when the fossil discovery rate increases for all the methods assessed (Tab. 3). The mean confidence interval widths obtained with the method Int are close to those obtained from the reference method Ref, again supporting the relevance of our way of dealing with the uncertainty on the speciation, extinction and fossilization rates. The mean confidence interval widths provided by methods S&S, McI, Alr and Glo are systematically greater or equal to those of methods Int and Ref. In particular, the mean confidence interval widths of method S&S are several times greater than those obtained with the reference method, notably when the fossil recovery rate is low. Namely, it is about 10 times greater when the fossil discovery rate is 0.05, and is still almost twice as large for a fossilization rate of 5 (Tab. 3). Note that, for the fossilization rates 1 and 5, the error percentages obtained with the method of Strauss & Sadler (1989) are not significantly lower than those obtained with methods Ref and Int (Tab. 2), despite the fact that its mean confidence interval widths are larger than those of these two methods. The same behavior is observed for methods McI, Alr and Glo for the fossilization rates 0.005 and 0.01. Though both methods McI and Glo have mean confidence interval widths very close to that of the reference for the two highest fossilization rates, only method Glo performs as well as methods Ref and Int for these fossilization rates, since the corresponding error percentages of method McI are higher than required.

Tables 2 and 3 show that method Glo performs systematically better than method McI. Let us recall that the only difference between these two approaches is the way in which the fossilization rate is estimated: branch by branch for McI, versus on the whole tree for Glo. This suggests that, in the case where the fossil process can be assumed to be homogeneous through the tree, a global estimation does improve the accuracy of extinction time estimation.

Method Int performs almost as well as the reference method Ref in the sense that both its error percentages and its mean confidence interval widths are very close to those of Ref, which are the best achievable under the assumptions used to simulate the data. The performance of method Glo is almost as good as that of methods Int and Ref only for the fossilization rates 1 and 5 (and slightly worse with the fossilization rate 0.1). Roughly speaking, method Glo differs from method Int in the sense that it does not take into account the diversification process (both methods assume that the fossilization follows a Poisson process). The respective performances of Int and Glo suggest that taking into account the diversification in the extinction time estimation does matter in the case where the fossilization rate is lower than the diversification rates but may be not essential when the fossil recovery rate is high.

Section S5 displays the results of a simulation study assessing the 50% and 75% confidence upper bounds provided by all the methods considered here. We observe the same general behavior as with the 95% confidence upper bound.

Results for the empirical example: extinction times for ophiacodontids, edaphosaurids and sphenacodontids

The tree showing the probability density distributions of extinction times of the tips shows that many lineages of all three clades were probably extinct well before the Kungurian/Roadian boundary (by considering only the tree displayed in Fig. 3). The extinctions seem to be spread out throughout the Kungurian, rather than being concentrated at any given time. These individual extinction times were also computed by taking into account the hundred trees of our dataset and are shown in greater detail in Figs. 4–7. This shows, unsurprisingly, that for nearly all taxa, the peak extinction time probability is well before the Kungurian/Roadian boundary, and that the 95% confidence interval does not reach that boundary, in most cases. One obvious exception is Dimetrodon angelensis, the most recent taxon included in this study (from the San Angelo Formation of the Pease River Group). Its peak density is near the Kungurian/Roadian boundary, and the tail of the distribution suggests that it could have become extinct well into the Guadalupian, although the posterior probability of the most recent fossil age of D. angelensis, biased towards the lower bounds of its interval, could be consistent with an older extinction time (Fig. S4). Despite this exception, our first hypothesis is confirmed; most lineages of the three clades appear to have become extinct before the end of the Kungurian. These results provide strong support for the first of our three hypotheses (that many ophiacodontids, edaphosaurids, and sphenacodontids had become extinct well before the end of the Kungurian).

Figure 3 One of the 100 equally parsimonious trees of our dataset and the extinction time probability density distributions of its extinct taxa.

Intervals of possible ages for each fossil are represented as thick brown segments with a certain level of transparency (darker brown segments correspond to over lapping intervals). The extinction time probability distributions are represented in red. These sometimes overlap the intervals of possible ages of the last fossils of a given branch because fossil ages are sampled in these intervals to compute the distributions. The thin blue line represents the Kungurian/Roadian (Cisuralian/Guadalupian) boundary. Only the branch tips are time-calibrated; the position of nodes is set to the median of the distribution of the corresponding speciation time, as shown in Figs. SI and S2 (Sections SI–S5).

Figure 4 Extinction time probability density distributions of ophiacodontids (X axis, in Ma).

The colored part under each distribution starts at its 95% confidence upper bound. The thin vertical lines represent the Kungurian/Roadian (Cisuralian/Guadalupian) boundary.

Figure 5 Extinction time probability density distributions of edaphosaurids (X axis, in Ma).

The colored part under each distribution starts at its 95% confidence upper bound. The thin vertical lines represents the Kungurian/Roadian (Cisuralian/Guadalupian) boundary.

Figure 6 Extinction time probability density distributions of sphenacodontids (X axis, in Ma).

The colored part under each distribution starts at its 95% confidence upper bound. The thin vertical lines represent the Kungurian/Roadian (Cisuralian/Guadalupian) boundary.

Figure 7 Extinction time probability density distributions of the taxa of our dataset that are not ophiacodontids, edaphosaurids or sphenacodontids (X axis, in Ma).

The colored part under each distribution starts at its 95% confidence upper bound. The thin vertical lines represents the Kungurian/Roadian (Cisuralian/Guadalupian) boundary.

Our results show that the taxa Ophiacodontidae and Edaphosauridae were probably extinct by the end of the Kungurian, with probabilities that these two clades persisted into the Roadian of only 9.1% and 8.0% for these two clades, respectively; this is also shown graphically by the fact that the 95% confidence interval on their extinction time only extends to the mid-Roadian, approximately (Fig. 8). Sphenacodontidae may well have become extinct later because the 95% confidence interval on its extinction time extends into the earliest Capitanian (Fig. 8). The probability that Sphenacodontidae extended into the Roadian is high, at 86.8%. In addition, the peak probability of its extinction time is near the mid-Roadian, significantly later than for Ophiacodontidae and Edaphosauridae, whose peak density of extinction is in the early to mid-Kungurian. Thus, our results provide weak support for our second hypothesis; although the probability that Ophiacodontidae and Edaphosauridae were extinct by the end of the Kungurian is high, the probability that Sphenacodontidae persisted into the Roadian appears fairly high. Note that these results consider the probable extinction times of unobserved lineages of these three clades, in addition to all those observed in the fossil record.

Figure 8 Probability density distributions of the extinction times (X axis, in Ma) of the clades Ophiacodontidae, Edaphosauridae and and Sphenacodontidae based on our dataset.

The colored part under each distribution starts at its 95% confidence upper bound. The vertical line represents the Kungurian/Roadian (Cisuralian/Guadalupian) boundary.

Contrary to previous suggestions (Lucas, 2018, p. 430) Ophiacodontidae does not appear to have become extinct before Edaphosauridae; in fact our computations suggest the reverse, but the difference in timing is not statistically significant (the probability that Ophiacodon tidae became extinct after Edaphosauridae is 0.533), so these extinctions should provisionally be considered more or less simultaneous. However, the difference between extinction time of Sphenadocondidae and that of the other two clades is marginally significant; the probability that Sphenacodontidae became extinct after Ophiacodontidae is 0.953, and for Edaphosaurida, this is 0.958. Thus, we cannot confirm the third hypothesis; in fact, our results suggest that Ophiacodontidae may have become extinct slightly later than Edaphosauridae. However, one element of this hypothesis is confirmed: Sphenacodontidae does appear to have become extinct last.

The mean fossilization, speciation, and extinction rates are 0.2306, 0.1348 and 0.1352, respectively. These rates are slightly higher for speciation and extinction, but much higher for fossilization, than those reported for a more inclusive dataset of amniotes (Didier, Fau & Laurin, 2017). That study reported rates of fossilization, speciation, and extinction of 3.21 × 10−3, 9.59 × 10−2, and 9.49 × 10−2, respectively.

For all these results, topology appears to have only subtle effects on extinction ages, although the effect on nodal ages is slightly greater, which is to be expected given that topology and clade content are intimately linked.

Discussion

The simulation study shows that taking into account all lineages in a clade, as well as the diversification process rather than only the fossilization events of a single lineage leads to confidence upper bounds of extinction dates which are both tighter and more accurate than those obtained from previous approaches, notably that of Strauss & Sadler (1989), which was designed to assess the true local stratigraphic range of a lineage in a section. Moreover, our approach can determine confidence upper bounds of the extinction time of taxa that have left very few or even a single fossil, a situation that is problematic for the methods that deal with each branch independently.

Comparisons between extinction times of individual ophiacodontid, edaphosaurid and sphenacodontid species (Figs. 4–6) suggest that lineages of these three clades became extinct at various times, which is more consistent with a long, low-intensity crisis than a brief, catastrophic event. These lineage-specific extinction times do not suggest a discrete extinction event associated with the Artinskian/Kungurian boundary, but they are compatible with (without proving) a mild, diffuse crisis spanning much of the Artinskian and Kungurian, as suggested by Benton (1985, 1989). These findings rather support the first of the three hypotheses that we test here.

The finding that one of three clades studied here (Sphenacodontidae) has a fairly high probability of having become extinct in the Guadalupian may seem surprising at first sight, given the few records (two occurrences of Dimetrodon angelensis) that may occur after the Kungurian (these may be either late Kungurian or early Roadian). Even for Edaphosauridae and Ophiacodontidae, an extinction in the early Roadian cannot be ruled out because the 95% confidence interval of their stratigraphic range extends into the mid-Roadian.

The extinction time of the most recent lineages of each clade and the extinction of the three large clades to which they belong (Fig. 8) reveals only small differences, except for Sphenacodontidae. This suggests that in these cases, the unobserved lineages (without a fossil record) of these clades did not add much to their stratigraphic extension toward the present. This probably reflects our increased sampling of fossiliferous horizons, which results in a much greater estimated fossilization rate. Didier, Fau & Laurin (2017, p. 981) estimated that “only about 14% of the Permo-Carboniferous eupelycosaur lineages (defined as an internode on the tree) have left a fossil record.” Using the new rate estimates, this proportion increases to 46%, which implies far fewer missing lineages. It is no surprise that for Sphenacodontidae, unobserved lineages seem to have a greater impact on the extinction time probability density, given that this is the most speciose of the three clades. The peak probability of extinction of Sphenacodontidae in the mid- Roadian is congruent with the conclusion of Brocklehurst, Kammerer & Fröbisch (2013). We obtained these results despite being fairly conservative in our assessment of the age of the San Angelo Formation; the range of possible ages that we assigned to it (from 275 to 271 Ma) extends from late Kungurian to early Roadian, but about two-thirds of this interval is in the Kungurian, and only a third in the Roadian.

Our interpretation of these results depends on the quality of the fossil record, which has to be of sufficient quality, on a worldwide basis (records in one location can compensate to an extent gaps in another) to meet the assumptions of our method, and incomplete enough to be compatible with the persistence of unobserved taxa for several million years. In other words, if the fossil record were extremely patchy, no method, no matter how sophisticated, would be able to extract well-constrained, reliable extinction dates from it. Conversely, if the fossil record were nearly complete, there would be no need for analyses because the history of taxa could be directly read in rocks. For studies such as ours (and those cited above) to make sense, an incomplete but not hopeless fossil record is required.

Is the fossil record sufficient to apply our method? Two kinds of gaps (temporal and geographic) can be problematic. In south-western North America, where synapsids have an excellent fossil record in the Cisuralian (as shown by our data), the continental fossil record of synapsids is poor in the Roadian, where it is restricted to the Chickasha formation of Oklahoma, and probably, by the San Angelo Formation in Texas. It has even been claimed that there is no Roadian fossil record of synapsids in North America because these formations have been argued to be Kungurian (Lucas, 2018), but our literature review shows otherwise (Clapham, 1970; Reisz & Laurin, 2001; Reisz & Laurin, 2002; Nelson & Hook, 2005; Foster et al., 2014; Soreghan et al., 2015). In any case, the North American strata that yielded Permian synapsids were deposited close to the paleo-equator. Recent studies show that there is a Roadian synapsid fossil record in Russia, given that the Kazanian is equivalent to the Roadian (Davydov et al., 2018). From the Wordian on, the synapsid fossil record is more widespread and is especially dense in the Karoo basin in South Africa (Rubidge & Day, 2020), in addition to Russia. These more recent strata were deposited in higher paleo-latitudes of about 25–30 degrees North for the Russian European Platform, and nearly 60 degrees South for the Karoo Basin (Schneider et al., 2020). This spatio-temporally disjunct synapsid fossil record has previously been commented by Brocklehurst, Kammerer & Fröbisch (2013).

To what extent can these geographic gaps in the synapsid fossil record invalidate our analyses? If Permo-Carboniferous synapsids were restricted to North America, where their fossil record stops in the Roadian (and probably, early in the Roadian), this would indeed be problematic. However, caseids extend into the Roadian (Maddin, Sidor & Reisz, 2008; Golubev, 2015) and varanopids survived until the Capitanian (Modesto et al., 2011), and both are documented by a sparse fossil record after the Cisuralian, and mostly outside North America. Even in the Cisuralian, caseids and varanopids have a poorer fossil record than ophiacodontids, edaphosaurids and sphenacodontids. This was noticed long ago by Olson (1965, 1968, 1975), who postulated that these taxa lived in a somewhat more upland environment than other Permo-Carboniferous synapsids. This hypothesized difference in habitat still seems plausible (Angielczyk & Kammerer, 2018, p. 130) even though Lambertz et al. (2016) postulated an aquatic lifestyle for at least the geologically most recent caseids, which have highly cancellous bone. The fossil record is inherently biased in favor of aquatic taxa (Shipman, 1981) because most fossiliferous deposits were deposited under water (Brett, 1995). Thus, the presence of a fossil record of caseids and varanopids in the Guadalupian suggests that the absence of ophiacodontids, edaphosaurids and sphenacodontids from that fossil record, despite their inferred habitat (apparently closer to the water than the areas inhabited by caseids and varanopids), is real, rather than a taphonomic artefact. This conclusion is also supported by the fact that ophiacodontids, edaphosaurids and sphenacodontids are also known from Europe (Berman et al., 2001), where amniotes have a Guadalupian and Lopingian fossil record (Schneider et al., 2020). This conclusion is further supported by the association of various combinations of these taxa (caseids and/or varanopids from Olson’s caseid chronofauna, with ophiacodontids, edaphosaurids and/or sphenacodontids, from another chronofauna that apparently inhabited more low-land environments) in some early Permian localities, such as El Cobre Canyon (Berman, Henrici & Lucas, 2015), the Archer City Bonebed (Sander, 1989; Reisz, Laurin & Marjanović, 2010), Fort Sill (MacDougall et al., 2017) and Bromacker (Berman et al., 2014). Thus, the absence of ophiacodontids, edaphosaurids and sphenacodontids from Guadalupian strata of Russia and South Africa suggests that they became extinct no later than in the Guadalupian (early in the Guadalupian, in the case of ophiacodontids and edaphosaurids).

The temporal heterogeneities in the quality of the fossil record appear to be less problematic, for Paleozoic synapsids, than the geographic bias, to the extent that there does not appear to be a trend of decreasing record. The Roadian appears to be a time of relatively poor fossil record of amniotes, but this record improves strongly in the Wordian (Olroyd & Sidor, 2017), which suggests that there is no decreasing trend in the quality of the fossil record of Permian synapsids. The assumption of a homogeneous fossilization rate through time, which is assumed by most methods (including ours) to study extinction events, does not imply that an approximately constant number of synapsid taxa (or specimens) should be known in each geological stage because the clade diversified through time and was affected by extinction events. We share the reservations expressed by Benton et al. (2011) about using the number of amniote-bearing localities to assess fluctuations in the quality of the synapsid fossil record through time (e.g., Brocklehurst, Kammerer & Fröbisch, 2013) because synapsids form half of amniotes from a cladistic perspective, so amniote diversification should be strongly correlated to synapsid diversification over time. This is problematic because Marjanović & Laurin (2008) showed that a simple exponential diversification model of lissamphibians best explained the variations in their fossil record, whereas the area of rock exposures of various ages played a negligible role. It is thus not surprising that Brocklehurst, Kammerer & Fröbisch (2013, p. 486) found that the “residual diversity” method conflicted with their two other metrics (species counts and phylogeny-corrected counts) in suggesting that synapsid diversity had decreased between the Wordian and the Capitanian, when other methods indicate a climax in synapsid diversity. To these problems in assessing biases in our assessment of paleobiodiversity, we must add the anthropogenic bias, which consists in the uneven effort that has been spent looking for fossils in strata of various ages, as pointed out by Brocklehurst, Kammerer & Fröbisch (2013, p. 481). Thus, assessing objectively the quality of the fossil record remains a challenging problem, especially in the terms relevant to our method, namely the probability of discovery of fossiliferous horizons per lineage per million years of those lineages that existed (most of which are probably not known).

Is the Roadian synapsid fossil record incomplete enough to be compatible with the persistence of unobserved taxa (Ophiacodontidae, Edaphosauridae and Sphenacodontidae) for a few million years? This seems likely. The recent discovery of a late Permian diadectomorph from China (Liu & Bever, 2015) serves as a reminder that the fossil record can still yield surprising discoveries that refute a long-established consensus about the stratigraphic range of taxa; this taxon was previously thought to have become extinct toward the end of the early Permian (Laurin, 2015). The recent discovery of two or three dinocephalian skulls that extend the stratigraphic distribution of this taxon (previously considered typical of the Tapinocephalus AZ) into the lowermost Poortjie Member of the Teekloof Formation also shows that even in a densely-studied, highly fossiliferous basin such as the Karoo, the time of extinction of a higher taxon (Dinocephalia is typically ranked as a suborder and considered to include three families) is not immune to revision (Day et al., 2015a). Our method could be used to assess the probability and magnitude of potential stratigraphic range extensions of taxa of various sizes, from single lineages to large clades, in addition to providing an additional statistical tool to better assess the evolution of biodiversity over time.

Conclusion

Our findings improve our understanding of the replacement of Permo-Carboniferous synapsids by therapsids in the Guadalupian. Therapsids must have originated in the late Carboniferous, as implied by their sister-group relationships with sphenacodontids (Reisz, 1986; Gauthier, Kluge & Rowe, 1988; Sidor, 2001; Benson, 2012; Didier & Laurin, 2020) and by our new dating of the eupelycosaur evolutionary radiation (Fig. S2), and more specifically of the sphenacodontid/therapsid divergence, which probably took place in the Gzhelian (Fig. S3). Angielczyk & Kammerer (2018) suggested that this event took place even earlier, in the Kasimovian, but this depends on the poorly constrained age of the Sangre de Cristo Formation in Colorado. The formation by the same name in New Mexico is from another basin, but a recent study concluded that it is of early Permian age (Lucas et al., 2015). Thus, there is currently no strong evidence that the sphenacodontid/therapsid divergence is older than Gzhelian. Yet, therapsids are unknown so far in the Carboniferous, with the possible exception of the very fragmentary remains (a string of a few vertebrae) from the Moscovian of Nova Scotia that Spindler (2014) interpreted as a therapsid, an interpretation that seems tenuous at best; the original interpretation that these belong to a sphenacodontid seems plausible (Reisz, 1972). Therapsids may be represented by a single specimen of Tetraceratops in the Kungurian, even though its affinities are still debated (Amson & Laurin, 2011; Spindler, 2020), and beyond the scope of this study. The first undisputed therapsids (dinocephalians from the Goluysherma Assemblage of the Russian Platform) date from the Roadian (Ivakhnenko, 2003; Golubev, 2015), but they become abundant only toward the end of the Roadian (Brocklehurst, Kammerer & Fröbisch, 2013, p. 487) or in the early Wordian. Lozovsky (2005, p. 182) reported that therapsids barely make up about 5% of the Mezen (Roadian) fauna and that the obvious domination of therapsids only occurred in the Tatarian (which started in the Wordian), as preserved in the Ocher and Isheevo localities. According to Golubev (2015, fig. 2), the Ocher subassemblage is early Wordian and Isheevo extends from the late Wordian through the mid-Capitanian. Also, some fragmentary Roadian caseid remains had been misinterpreted as therapsids (Brocklehurst & Fröbisch, 2017). Thus, in the early Roadian, the replacement of Permo-Carboniferous clades of synapsids by therapsids had apparently barely started (Brocklehurst, Kammerer & Fröbisch, 2013). The dynamics of biodiversity changes in synapsids similarly shows that therapsids surpassed other synapsid clades in biodiversity sometime in the Roadian (Brocklehurst, Kammerer & Fröbisch, 2013, fig. 1). All this suggests a very slow initial therapsid diversification (but this could be tested more rigorously with the method presented here), in contrast with the explosive diversification model postulated by Kemp (2009). But was the replacement of Permo-Carboniferous synapsids by therapsids competitive or not? Our findings shed new light on this question.

The extinction of Ophiacodontidae and Edaphosauridae near the Kungurian/Roadian boundary supports the hypothesis, recently proposed by Olroyd & Sidor (2017), that the replacement of Permo-Carboniferous synapsids by therapsids was non-competitive because the absence of definite therapsids in the Kungurian fossil record suggests that they remained rare at that time, at least locally. It is thus difficult to envision that therapsids played a major role in the extinction of Ophiacodontidae and Edaphosauridae. The slightly later extinction date of Sphenacodontidae gives a more ambiguous signal on this point. They may also have died out before therapsids had become major competitors, as a literal reading of the Roadian fossil record suggests, but this is less certain. It is conceivable that the replacement of Permo-Carboniferous synapsids by therapsids was partly competitive (for sphenacodontids, and more likely, for caseids and varanopids, which are known to extend at least into the Roadian and Capitanian, respectively), and partly opportunistic, by filling the ecological vacuum left by the extinction of ophiacodontids and edaphosaurids. A dating analysis of the therapsid evolutionary radiation through the FBD would be helpful to assess this. Nevertheless, our findings suggest that the exinction peak in Permo-Carboniferous clades (Olson’s extinction) was near the Kungurian/Roadian boundary, rather than in the Roadian and Wordian, as suggested by Sahney & Benton (2008, p. 760).

Supplemental Information

Supplemental Information 1 Data, source code and R package.

Archive of the github repository

Click here for additional data file.

Supplemental Information 2 Supplementary information.

Click here for additional data file.

We thank Robert W. Hook (U. of Texas at Austin) for a wealth of information and discussions about biostratigraphy and lithostratigraphy, especially for the El Reno and Pease River groups, and for putting one of us (ML) in touch with other specialists of Permian stratigraphy. We also thank Gregory P. Wahlman (Wahlman Geological Services) and John R. Groves (Technology Center, Pittsburgh, PA) for information about Permian fusulinids. This draft was improved by comments from Neil Brocklehurst, Charles R. Marshall and Ken Angielczyk. Manon Thomazo helped with the compilation of character and state names in the Mesquite Nexus file (see supplements) and corrected some typographic and stylistic errors in parts of the draft.

Additional Information and Declarations

Competing Interests

Author Contributions

Data Availability

The authors declare that they have no competing interests.

Gilles Didier conceived and designed the experiments, performed the experiments, analyzed the data, prepared figures and/or tables, authored or reviewed drafts of the paper, and approved the final draft.

Michel Laurin conceived and designed the experiments, performed the experiments, analyzed the data, prepared figures and/or tables, authored or reviewed drafts of the paper, and approved the final draft.

The following information was supplied regarding data availability:

The source code, R package and data are available at GitHub:

https://github.com/gilles-didier/DateFBD.

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
