# Peer review of "Distributions of extinction times from fossil ages and tree topologies: the example of mid-Permian synapsid extinctions"

_PeerJ, doi:10.7717/peerj.12577_

## Round 0.1 · original submission · Major Revisions

· Academic Editor

Major Revisions

The paper is very clearly written and I'm very much confident it'll be a great addition to the literature on FBD models.

Having said that, I have now received back three thorough reviews on your paper. All of them were very positive, but also pointed out some aspects that could be improved. R1 and R3 suggest you compare your model with other ones, instead of only Strauss & Sadler. They also points out that some claims seem unsupported. Consider re-writing or suppressing them. R2 suggests a number of instances in the introduction to be improved and asks for moderation in some arguments.

Even though changes seem substantial, I believe they won't be difficult to implement. But I'm recommending major revisions to give authors more time to work on the manuscript.

·

Basic reporting

On the whole I think the paper is excellently written, with a clear background provided and figures and data all clear. The paper is by necessity very equation-heavy, but I think they do a fine job of supplementing this detail with an explaination of exactly what is being output by the analysis. There are a couple of points where I think things need to be clarified, and a couple of statements that I think are unsupported (see general comments below), but on the whole the basic reporting is excellent

Experimental design

On the whole the method seems sound. The R package is uploaded in github with instructions on how to run through the analyses. However I would suggest that an R script, raw data and isntructions also be uploaded as a supplement, purely for purposes of repeatability; the actual analysis run in this paper should be set in stone, and github be used to track updates and advances on the method.

I do have some issues with the simulation study and also the phylogenetic framework used in the study (see general comments below) but these should be fairly easy to resolve.

Validity of the findings

As I stated above, I do have some issues with the simulation study and the phylogenetic framework employed, but these shouldn't be too horrendous to resolve.

Additional comments

This paper represents a useful contribution to the array of methods for studying extinction in the fossil record. The authors have put a lot of thought into the study design, and also present a useful empirical example with implications for early tetrapod evolution.

On the whole I strongly recommend this paper be published, and I think it will represent a useful framework for future macroevolutionary analyses. I do have a few suggestions for both the empirical and simulation studies, and also think there are some points where statements need further support or clarification before publication.

For the empirical studies, I think the phylogenetic framework used could do with a few updates. I know the tree figured is an illustrative one drawn from several analyzed, but some of = the relationships shown are contradicted by recent updates to pelycosaur phylogeny and should not have been included in analysed trees. Ones that leapt out at me are: that Milosaurus is likely not an ophiacodontid (Brocklehurst & Frobisch 2018, Journal of vertebrate Paleontology volume 38); some rejigging required to the sphenacodontid phylogeny based on the most recent assessment (Brocklehurst & Brink 2017), including that Sphenacodon, Ctenorhachis and Ctenospondylus form a clade; not sure why Echinerpeton has been allowed to form a closer relationship with Varanosaurus than with Ophiacodon, when in all analyses that have included it it has formed an outgroup to a clade containing Varanosaurus and Ophiacodon (see updates to the Benson analysis, in particular from Mann & Patterson 2020, Journal of Systematic Paleontology volume 17).

Another idea that the authors may like to try (but I'd consider this not compulsory) is, instead of drawing equally parsimonious trees for analysis, and then choosing one at random for illustration, would be to resolve polytomies via analysis with the fossilised birth death model, constraining the tree in the same way as has been done before, and running an FBD analysis using observed tip age estimates, would resolve the polytomies based on the age priors and other FBD parameters. The analyses could be carried out on randomly drawn trees from the posterior and the maximum clade credibility tree could form the basis of the figures

For the simulation studies, the authors demonstrate that their method outperforms the Strauss & Sadler method, the oldest method of the family they examine. But I do wonder why they limit their comparison to just that method. After all there have been a whole host of expansions on that theme, as well as some methods that have used the concept but taken a new approach. I am particularly thinking of the Creeping-Shadow-of-a-doubt method (Alroy 2014, Paleobiology volume 40) and GRIWM (Bradshaw et al 2012, Quaternary Science reviews volume 33); although the former has not been widely used, the latter has been rather popular particularly in studies of the Pleistocene extinction. I think it would be useful to see how your method compares to other advances that have been made since 1989.

Some more specific line by line comments:

Line 56-60: “A third problem of the classical taxic approach is that the known stratigraphic range of taxa typically underestimates their true stratigraphic range (real age of appearance and extinction), a problem that is likely to be especially acute for taxa with a poor fossil record (Strauss and Sadler, 1989). Most recent analyses using the taxic approach still rely on a fairly literal reading of the fossil record to the extent that they use first and last stratigraphic occurrence data as if it were the true stratigraphic range of taxa.”
- I'm not sure the authors are being entirely fair here, particularly as some of the studies cited as using the classic taxonomic approach on lines 41-42 employ sampling corrected estimates of extinction (e.g. Brocklehurst 2018 uses the second for third correction). Such methods don’t endeavor to extend the ranges of specific taxa, it is true, but they definitely don’t make the assumption that the observed origination and extinction of a taxon is true, rather they seek to correct for the gappiness of the fossil record by adjusting the estimates up or down depending on how incomplete the fossil record of that time is. I would frame the issue as your method places confidence intervals on the extinction extimate for specific taxa, not that other studies of extinction have assumed that the observed extinction time is true (if you read most of the papers you cite on lines 41 and 42, it is clear most of them absolutly do not assume this; they just account for it as a general sampling issue rather than endevouring to resolve it for each specific taxon).

Line 156: “A recent literature review of Middle Permian amniote-bearing localities (Olroyd and Sidor, 2017) shows that the assumption of a reasonably homogeneous fossilization rate (worldwide) through the studied time interval (Bashkirian-Roadian) is reasonable”
This is an absolutely bizarre statement that I really cannot support. First, I have had a quick look at the paper cited and I can't find anywhere a suggestion anything like that! Even if there is something I’m missing, can you really take a statement like that from a review of the Middle Permian, and assume it applies from the Pennsylvanian through to the earliest middle Permian? I really doubt it. The Carboniferous and early Permian consist of preservation modes ranging from Texas red bed formations to preservation within fossilised tree truks to cave deposits. To assume that fossilisation potential and rate is homogenous across these preservation modes flies in the face of common sense, as well as actual study of preservation (e.g. Texas red beds selectively preserve larger taxa [Behrensmeyer 1988, Palaeogeography, Palaeoclimatology, Palaeoecology volume 63] while Richards spur has a mode of preservation better suited to smaller taxa [MacDougal et al. 2017, Palaeogeography, Palaeoclimatology, Palaeoecology volume 475]). I don’t think the variation in fossilisation rate is a fatal flaw that will invalidate the study; after all it applies to most methods. Why not just acknowledge that it’s a widely applicable bias that is not accounted for, rather than endeavor to justify it in this rather spurious way?

Line 369-370: “The computation presented above allows us to determine the distribution of the extinction time of a subset of taxa in the case where where the tree topology.” The wording is ambiguous as to whether "distribution" meant a range of exact extinction estimates of each taxon in the subset, or a probability distribution of extinction estimates. I assume the latter so maybe specify a probability distribution. Also note the word “where” has been repeated

Line 511: "The finding that one of three clades studied here (Sphenacodontidae) has a fairly high probability of having become extinct in the Guadalupian may seem surprising at first sight, given the lack of records after the Kungurian."
- In their discussion of this surprising result, no specific mention is made (just rather vague hints) that the fossil record from North America and Europe, where Sphenacodontidae were found, is not present beyond their last appearance. So if the confidence intervals suggest they survived beyond their last appearance, that seems the most likely explanation.
- This point did draw another to my attention: from their discussion and the figures I infer that the authors are using ages of the formations consistent with those advocated by Lucas, rather than the Roadian age for the Chickasha and San Angelo formation advocated by eg Reisz & Laurin 2001, Benton 2012 Geology volume 40, Brocklehurst 2020 Proceedings of the Royal Society B volume 287. Maybe good to state this explicitly, as this does impact on your discussion regarding competition between therapsids and pelycosaurs e.g. line 546: "The dynamics of biodiversity changes in synapsids similarly shows that therapsids surpassed geologically older synapsid clades in biodiversity sometime in the Roadian (Brocklehurst et al., 2013, fig. 1)." the study cited uses the Roadian age for these formations, so there was an observed interval of temporal overlap between the therapsid-bearing formations of Russia and the pelycosaurs-bearing formations of North America.

Line 503: "compatible with (without proving) a mild, diffuse crisis spanning much of the Artinskian and Kungurian, as suggested by Benton (1985, 1989) and by Brocklehurst et al. (2013)." Don't think Brocklehurst et al. 2013 should be cited here. Its been a while, but I'm fairly sure I never suggested even a mild crisis spanning the Artinskian and Kungurian; rather I suggested the Artinskian was an interval of recovery from a mild crisis at the end of the Sakmarian, and the crisis was concentrated in the late Kungurian (also what I suggested in Brocklehurst 2018, cited in your work).

Figure 3: I know you acknowledge that the node ages of the phylogeny are not accurate, and I know the node ages are not relevant for the inferences of extinction. A phylogeny with accurate node ages is shown in another figure. All of which makes me wonder what the point of this figure is? Why not put the information shown (the probability distribution of the extinction estimates) on a figure with accurate node ages? This figure will just cause confusion.

With these changes made, I think this study will represent an excellent contribution

Neil Brocklehurst

·

Basic reporting

See comments.

Experimental design

See comments.

Validity of the findings

See comments.

Additional comments

This is an excellent manuscript, presenting a new method for accommodating the incompleteness of the fossil record when trying to estimate time of extinction (and origination). I have no doubts that it should be published in PeerJ. The method is able to make better use of the data than current methods, with the sacrifice being the need for assuming constant preservation rate across all lineages (which the standard methods, those based on Strauss and Sadler [1989], (S.S.) do not do). But as one of the strongest proponents of the S.S. set of methods, I find the method proposed here very appealing.

Below I have a set of suggestions for improvement. Most are minor, but I think the author need to pay particular attention to:

#10, #12, #46, #47 (what really makes their method perform better than S.S.; I don’t think it is the reason they state in the abstract, or at least that is just part of the reason)
#25 (sensitivity to different tree topologies)
#26, #39b (we need the temporal data to be presented somewhere, and ideally the rational for the age calls and their ranges).
#34–36 (effect of only taking simulations with > 20 taxa on the actual preservation rates simulated)
#39 (increasing the clarity of Figure 3 (and 7).
#41 (we need more data to have confidence in the claim of a no end-Kungurian extinction, given the insidious role of hiatuses in the rock record)
#43 (how to compute extinction density for clades)

And then also I have suggestions for a more balanced introduction (see #5–10).

1. Line 13. The abstract says that the method is designed for a tree with extinct and extant taxa, but in the end their example has no extant taxa (which is not a problem). I feel the sentence is a little bit of a ‘mis-direct’, in that red flags go off in my head when I see the inclusion of both extant and extinct taxa, because typically the extant taxa outnumber the extinct in numbers and data completeness, and thus can suffer from data imbalance where the Recent gets undue weight (this is a problem with other methods that incorporate fossils, in that they are still pretty biased, for example). So, this rather long comment, in the end, amounts to the recommendation that the authors say something like “Given a phylogenetic tree of just extinct, or extinct and extant, or just extant taxa, where at least some fossil data are available, we present [I would not use ‘devise’, but this is largely a matter of taste] ….
2. Line 16. Is your method also unusual in that it can incorporate taxa known from just one fossil (this is not unusual for FBD, but it is for confidence intervals). So maybe this part of the abstract needs to be a little more nuanced (in that ‘previous ones’ include disparate methods that ‘bite down’ on the data in different ways). Maybe the simple solution is to simply be explicit about what you mean by ‘’previous one” (line 16).
3. Line 23. Virtually no one is going to know the Permian stage names (I don’t, and I am a paleontologist), so give us some help here, e.g., say something like Cisuralian (Mid-Permian, or whatever it is), and then, indicate that the next interval is younger in some way (you could also give approximate age in millions of years, that might make all the difference).
4. Line 29: Suggested rewording: “ … direct evidence (even if fragmentary) of the ….”
5. Lines 44-45: I know opinions differ, but I find Patterson and Smith flawed because it failed to take into account that the extinction of a paraphyletic group, does in the end, often (granted not always) equate the extinction of typically several lower-level taxa – so for example, P&S would have no end-Cretaceous extinction for fish or echinoids, which is just silly. So, the point about paraphyly is of real concern, I agree, but I would urge the authors to be a little more nuanced here (especially as this issue has no bearing on the method they present).
6. Lines 52-55. Again I would urge moderation/nuance – higher taxa often represent group of species with similar morphologies, and thus similar ecological roles, so the diversity dynamics of higher taxa capture a different signal than species-level analyses, and can yield insights not possible with species level data (to take an extreme case, a species-only level analysis would not pick up the Cambrian explosion, or at least only see it as a pretty small burst of speciation, dwarfed by the Ordovician for example). But the lack of intrinsic comparability for, say families, is real, so I am not disagreeing with the problems the authors identify (and Foote’s Ph.D. was based on trying to circumvent this by developing disparity approaches to deal with the issue).
7. Lines 56-60. Again, a little bit of moderation would be good. For example, Foote (2003, J. Geol. 111:125) and Lu et al. (2006, PNAS 103: 2736) provide analyses of Phanerozoic marine biodiversity dynamics with corrections for the incompleteness of the fossil record. But the point is well taken.
8. Lines 61-69. This paragraph mixes two things that need to be dissected. First is the problem of low temporal resolution of binned data (typical of compilations), but again Foote has tried to deal with this (see his 2003 paper, referred to above). Then there is the problem of Signor-Lipps when you actually have ‘section’ data available. The only paper I know of that deals with the Signor-Lipps effect at the stage level is Lu et al. (2006, PNAS 103: 2736), who argue (and quantify) the effect at the stage level for Sepkoski’s data. So, there is a (small) literature here on this topic ….

I would add here that my comments 5–8 I think could be accommodated easily, without adding much text – in the end, these points are immaterial to the method presented, but a more balanced set of views might increase acceptance of the method.

9. Line 75. You might also want to add Tavaré et al. (2002, Nature 416: 726).
10. Lines 92-110. Intuitively, it is not clear to me that the primary advance of the method proposed here is its ability to take into account diversification after the LAD of a lineage prior to its extinction. Rather it seems to me that its ability to use all the fossil data (i.e., include lineages with just one fossil), and to use all the fossils together (at the cost of assuming a constant preservation rate), is the real strength. As a test, have the authors tried the sensitivity analysis of turning off speciation after the LADS and seeing if it makes any difference? Or is it simply that by allowing for post-LAD diversification, the method allows for that fact that the lineage might have lasted longer than one might have thought with the know fossil record. So, maybe the argument that allowing for post-LAD diversification does have an intuitive basis. I would encourage the authors to add in that intuition, whatever it turns out to be.
11. Lines 111-125. Agreed! This is an exciting paper!
12. Lines 126-134. Agreed! This is an exciting paper! I wonder if these two points might be more (or as) important as the possibility of post-LAD diversification, and so I would encourage the authors to think about this, and maybe bring these points up higher in the paper, for example, in the abstract.
13. Line 136. Both extinct and extant taxa are mentioned – please see my point #1 above.
14. Lines 142-146. Nice distinction.
15. Line 151. You say your method assumes ‘fairly homogeneous’ fossilization with time. It would seem that you actually assume a homogeneous rate, so delete ‘fairly’?
16. Lines 151–162. A potential red flag here. You comment that a relatively homogenous fossil recovery rate for your taxa, globally, seems reasonable. You then comment that you will not be following Lucas’ (2017) recommendation for a ‘best-basin’ style of analysis. OK, I can live with that. So, the big question for me is how did you correlate the sections to provide the resolution provided in your figures. I have always balked at applying Straus and Sadler on global data for the inability to line up the appropriate sections (but Shaw, or Sadler’s type methods could be used). So, I am sure hoping you deal with this appropriately – I will read on (but maybe let me know here that you will deal with this).
17. The review of the extinction scenarios in the literature is a long section, but I think is of value. However, it was almost too much for me, largely because I don’t know the stratigraphic nomenclature. Now, in the description of the first proposed extinction ‘event’ you give some absolute ages, which I found very useful. But then you don’t do so for the subsequent ones. So, simply, I would add absolute dates in the paragraphs that begin on lines 200, 221, and 241, as you have for the paragraph that begins on line 187, to help keep the reader oriented.
18. With respect to my comment #17, I wonder if a synoptic figure might help (but I can also see that there are enough variants of each of the proposed extinction scenarios that this might not be possible).
19. With respect to comments #17 and #18, I find myself wondering how many of the variant hypothesizes of extinction rejected by this paper, where rejected because of the method itself, and how many were simply rejected due to improvements of the correlation scheme which you presumably had to develop to apply the method. Dissecting out these two components might be useful for those who are interested in the method, and not the taxa (having said this I have now reached the paragraph on line 253, and this does a pretty good job of stepping back, so well done).
20. Line 272. Whenever I see cut-off like 95%, but with many taxa, I find myself reflexively wondering about whether a Bonferroni-type correction needs to be applied. Yes, no?
21. Line 272. I don’t know where the Kungurian fits in! I forget already if it relates the to 1st, or 4th of you reviewed extinction events (and given that you are not going to study them all, maybe you should say so, when you present them). So, help me, by refering to a synoptic figure (if you decide to add one), or something ….
22. Line 292. Valuable note – very good.
23. Line 300. Valuable note – very good.
24. Line 311. Looking forward to reading about your age assignments (see also comment #16).
25. Line 312. Whoa, 15 x 10^6 trees! And you picked just 100. Have you performed any sort of sensitivity analysis on the impact of the uncertain tree topology?? You have to deal with the range of potential topologies.
26. Lines 314-328. I agree with the reasoning for working with absolute time rather than stratigraphic thicknesses, for several reasons. So, no push back from me. However:
a. Can you add the age model and assigned ages you used in a Supplementary table, or whatever?
b. More importantly, what is the impact of the uncertainties?
c. How did you decide on the ages, and their potential ranges?
d. Did you count fossil horizons at individual localities? Or just give counts at the Formation level.
e. We need a proper description of these data.
27. Lines 330 and following. FYI, I have not evaluated the maths or description of the FBD model.
28. Lines 374-377. Is this what you did to deal with the temporal uncertainties? If so, please refer the reader when he/she is in lines 314-328 to the fact that they will be given an answer below.
29. Lines 354-355. For a paleontologist, this is cool. I wonder if this capacity should be referred to in the abstract?
30. Lines 409-410. The units for these rates? Be explicit (orig/ext per lineage per million years, what?).
31. Lines 409-410. So, you average longevity is 1/0.19 myr = ~5 million years? Does that make sense with what you know about you taxa. Maybe just mention the expected lineage duration with this extinction rate, so people can gain an intuitive feel for the nature of the simulations
32. Line 422 vs. line 424. I have probably missed something, but line 422 says 5,000 sets of rates, but line 424 says 1,000 simulated trees. Can you reconcile this (either fix, or explain in the text why they are not the same)?
33. Table 1. So, I should be dividing the total number of extinct taxa by 1,000 to get the number/tree? Maybe say so in the caption, directly.
34. Table 1 (and Tables 2 and 3). I think there is something amiss-misleading here. If I am reading this properly, the preservation rate of 0.005 is giving ~21 taxa/tree, while the preservation rate 1,000x larger (5) is only giving ~36 taxa/tree. How is this possible (not even double the number of extinct taxa despite the 1,000-fold increase in preservation rate)?! I think the reason might be simple – on line 412 you state that you only took trees with > 20 fossils. So, I presume that means that for the lower preservation rates (or most of them) you had to stochastically wait a long time for the very few trees that met your criteria. In which case, I bet the actual preservation rate (measured retrospectively on the chosen trees) for those few trees was way higher than the numbers presented in Table 1. Given the closeness of the values from the lowest to highest preservation rates (from 21 to 26 fossils/tree), it looks to me that the realized preservation rates are quite high and relatively uniform (at least in log-space), and that the reported fossilization rates are seriously misleading. Can you compute the realized fossilization rates? And if they are way higher than the simulated rates (as I suspect), then change the description in the text and presented data to reflect this point?
35. Table 1. Sorry for thinking out loud here ¬– but the %S.S. feasible column does show the expected trend, so I think another metric that would be of value for understanding the simulations is the total number of taxa (on average) in the simulations (or another way of saying this, what is the proportion of extinct taxa to total taxa). It looks like you go from mostly singletons for the low fossilization rate, to mostly non-singletons for the rate that is 1,000x higher. With respect to my comment #34, the %S.S. column is the most important, so I don't think the simulations need re-doing.
36. Lines 425-437. This may need a little re-writing considering comments #34 and #35. I think saturation is happening (what else could it be?). And that tells me that I can guess the size of your trees (because saturation means all taxa have fossils). But also says that your censoring of the initial simulations also means the reported preservation rate (r) is misleading, and thus the ratio of r to p and q.
37. Table 2. Nice demonstration that the method they propose does what it claims to do. Very good. As an aside the fact that the italics values are almost always bigger than the non-italics suggests to me that the data amenable to S.S. are biased in some subtle way, which is why those values indicate confidence intervals that are too long (but this is my just thinking through the data presented to see if they make sense, and they do).
38. Table 3. This is good. I think it is enough for this paper, but 95% confidence intervals are very long when you only have two fossils (19x the observed stratigraphic range). I would be interested to see what the unbiased estimates look like for the true time of extinction (median gap size), or, simpler, the 50% confidence intervals. The average of these should be 0 million years, roughly, with half above and half below the true times of extinction, if the method was working correctly. But the Table does the job it needs to, and so while I am curious about the 50% confidence intervals (or in your case, the middle of the density distribution) it is not needed here (BUT, if they want to use their method to actually estimate extinction times, then my suggestion might help).
39. Figure 3. Nice. But I have questions.
a. Why the various shades of brown on the lineages (i.e., why not just one color?)?"
b. For fossil occurrences I am expecting dots, but I am seeing ‘sausages’. Do these indicate the age uncertainties? If so, does that mean that the taxa with just one ‘sausage’ is only know from one fossil (or one formation, and you have not dissected out the fossils within that section?)? Do tell!
c. Pretty obviously the red bits are the extinction probabilities (please say so).
d. In several cases the peak density for extinction seems to lie well within the ‘sausage’ (for example, for the top taxon Stereorhachis). This does need some explanation (and may well be answered by your answers to (a)-(b)).
e. I know the caption says the scaling of the bottom part of has been extended for clarity, but when I first looked at the figure, I had not read the caption, and thought the tree was crazy; how can those nodes be so deep without at least some fossils!? So, maybe make the lines dashed, or light grey. Better, why not just use the branch lengths from Figure 7, but leave off the probability densities, given how noisy the figure would be if they were included (and then you can get rid of at least the Devonian part of the figure that serves no purpose).
f. Put “Million of years ago” on the x-axis, small print, at the lower left.
g. Given that you are going to talk about three higher level taxa next, why not indicate their membership on Fig. 3?
40. Figure 4. Make the lines a little thicker, perhaps.
41. Results. Hypothesis 1 (starting line 456): OK, nice, but a little simplistic in the interpretation. As we all know you can’t find fossils if there isn’t any rock. So, if there is a lack of outcrop at the top of the Kungurian, for example, then you can’t say much about the actual time of extinction of those taxa that you have shown disappear from the record before that boundary. I would like to see two things done before the authors claim a non-Kungurian extinction:
a. First, to help my own eye, it would be great if the taxa, or perhaps better, their probability density distributions were colored by their (coarse) geographic location. Why? Because if an a given region all (or most) of the taxa happened to survive into a time of no rock, then their inferred times of extinction would cluster about the time of the youngest available rock (so for example, I am suspicious about taxa 5, 6, [11], 21, 22, 29, 30, 40, [41], 42, [45], 46, 48, [49] – if they all came from the same geographic area, then the data look consistent with all of them disappearing from the fossil record at about the same time (which is interesting in its own right), and then if there is no younger rock (that goes to the Kungurian boundary) in that region, then you can’t say when they went extinct.
b. Some sort of statement about what we know about the continuity of the taphonomic window for these taxa across the stratigraphic intervals of interest.
42. Results. Hypothesis 2 (starting line 476): Maybe just say ‘fully extinct”, or something. I was having trouble with post Kungurian survival given the previous results. I guess what I am saying is that having the last occurrence of a whole clade much later the time of extinction of most of the constituent lower taxa is uncontroversial, but maybe you need to simply point this out, so a late extinction does not seem to contradict the earlier species-level extinctions.
43. Results. Hypothesis 1 (starting line 456): You need to describe (probably in the methods section) how you compute the extinction density for a clade. Do you use all the data for the clade (which would seem to severely violate the constant Poisson probability of finding fossils, given that the probability of finding a clade member is presumably proportional to the number of taxa present); or have you simply used the density of the single species with the latest potential extinction time; or have you multiplied the densities somehow? This needs to be explained (or if it is, give the block of text its own section/title).
44. Line 480-481. You say that the Sphe. have a peak density near the K/R boundary, “as for Oph. and Edaph.” I think this is just a typo – Oph. and Edaph seem to have much earlier peaks.
45. Line 483. You could also simply that there is a greater probability that the Sphe. survived the Kungurian than not ….
46. Line 494. See comment #10. I am not convinced that this is the reason the method performs better.
47. Line 506. Oh golly! You seem to actually agree with my skepticism expressed in comments #10 and #46! OK, you really need to change the relevant text throughout!
48. Paragraph beginning on line 511. Good!
49. Sentence beginning line 527. Marshall (1990, Paleobiology) suggested this (applying it to the platypus fossil record) for times of origin. And maybe here cite the papers referred to in comment #7.
50. Appendices. For the record, I have not worked through the Appendices.

·

Basic reporting

General Comments: This manuscript presents an extension of the fossilized birth death model that tries to account for potential unseen diversification after the last appearance of a taxon in the fossil record, and applies it to the question of when ‘pelycosaur’-grade synapsids went extinct. The latter topic is of interest because there is on-going debate about the timing of pelycosaur extinctions (i.e., whether there was a tetrapod extinction event near the early-middle Permian boundary) and its potential relationship to the diversification of therapsids. Overall, I found the paper well organized and thought-provoking. The extension of the FBD model seems like it will be of general interest, and I think that the paper makes a positive contribution to our picture of synapsid diversification/extinction dynamics. Most of my comments are pretty minor, but one weakness of the paper concerns citations. There are a number of relevant papers, mostly published in the last 3-5 years, that I think should be cited in various places in the manuscript. As it is, the citations in the paper make it seem like something that could have been written a few years ago, and I think the additional references will help it feel more up-to-date.

Experimental design

see below

Validity of the findings

see below

Additional comments

Line 32: I’m not sure I agree with the statement that changes in rates of cladogenesis and extinction generated evolution radiations and mass extinctions. I suppose in some sense that’s true (i.e., a radiation can’t occur without an increase in speciation rate), but it seems like something that is more associated with those types of events instead of the generating force. The generating force that causes the increases in rate seems like it would be something like environmental change, the emergence of a key innovation, etc.

Line 37: I think spectacularly is a little hyperbolic here. I recommend replacing with greatly.

Line 47: Maybe provide some examples of studies that proved him right.

Line 53: change to lineages

line 56: change to ranges; also change to underestimates in the next sentence

line 60: change to: “...data as if they were the true stratigraphic ranges...”

Line 78: These papers pretty much deal with just the Permo-Triassic extinction, so I recommend adding ‘e.g.’ at the start to indicate they are just some examples given that other extinction events also have received a lot of attention. Also, maybe consider citing Viglietti et al. (2021, PNAS) as the most recent work looking at extinction patterns in the Karoo Basin during the PTME.

Line 87: change to considered

Line 90: I recommend making the sentence starting ‘See’ into a parenthetical citation in the previous sentence.

Fig. 1: I recommend changing the color scheme of this figure to make it more accessible (i.e., avoid using red and green together).

Fig. 1. caption: Change to: “A simulated extinct clade with sampled fossils represented by brown dots. Top: The clade’s complete evolutionary history. Bottom: The portion of the clade’s history observable from the known fossil record. Note that the ‘blue’ and ‘green’ taxa diversify before going extinct, but that these diversification events are not recorded in the known fossil record.”

Line 108: replace this with the

Line 116: change to cladogenetic events

Line 117: Brocklehurst (2020; Proceedings B) might be good to cite here as an example of a study that used a FBD-approach to look at extinction in some sense. His approach and interest to the problem is different than yours, but the anaysis was still partially framed in an extinction context.

Line 124: Maybe cite a couple of Spencer Lucas’ tetrapod biochronology papers here as well, since they are classic examples of using higher taxa this way.

Line 132: change to: “...(see Simulation Study below).”

Line 135: change develop to extending

Line 136: change namely to specifically

Line 155: replace on with to

Line 158: A couple of thoughts here. First, I agree that your method makes the most sense when applied globally instead of locally. However, it seems almost certain that fossilization rates (and sampling rates by paleontologists) will be uneven in such cases. Even in the pelycosaur example you use here, that is very likely. For example, thousands of specimens of your focal taxa have been collected from the south western U.S. during this time interval, but many fewer are known from Europe. Second, I don’t think the Olroyd and Sidor study is really appropriate to justify your assumption of even fossilization rates for a few reasons. First, it deals with a different time interval (middle Permian vs. Pennsylvanian to early Permian). Second, it primarily deals with different geographic areas, given the shift in the main locations of tetrapod fossils from low latitudes to higher latitudes that takes place near the early-middle Permian boundary. Third, Olroyd and Sidor’s review shows that the middle Permian tetrapod record is quite unevenly sampled (e.g., many specimens known from areas such as Russia and South Africa, but very few from Brazil). Something like Brocklehurst et al. (2013) would probably be more relevant to cite, although the message of that paper may conflict with what you’re trying to say here.

Line 164: change to trees

Line 178: Because the epochs of the Permian have formal names (i.e., Cisuralian, Guadalupian, Lopingian), early/middle/late aren’t capitalized when talking about the Permian. This change should be made throughout the paper.

Line 181: I recommend citing Angielczyk and Kammerer (2018) here because that paper includes some discussion of the likely Carboniferous origin of therapsids as well. Spindler (2014) might also be good to cite since he tentatively re-identifies some Pennsylvanian material as representing therapsids.

Line 201: You definitely should cite Sahney and Benton (2008) here because it was really the start of the current discussion of ‘Olson’s extinction’. I also recommend citing Brocklehurst (2018) as well.

Line 220: Although he doesn’t consider Strauss and Sadler-style confidence intervals on stratigraphic ranges specifically, Brocklehurst (2018) is an attempt to address whether Lucas’ (2017) literal reading of the fossil record is likely to produce accurate results.

Line 221: change to event

Line 229: It’s a bit uncertain what you mean here by “about the same time.” Do you mean in Roadian (i.e., in reference to the previous sentence about Ennatosaurus) or the time of the end-Guadalupian extinction in general? More importantly, more recent studies have not found a decrease in therocephalian diversity at this time: see Huttenlocker and Smith (2017) and Grunert et al (2019). They disagree in the exact details of diversity increases at this time, but in general find evidence for increasing diversity across the Guadalupian-Lopingian boundary.

Lines 241-252: I think this paragraph needs to be updated to take into account the large amount of work that has been done on the terrestrial Permo-Triassic extinction (and ideally also work on the marine realm as well). For example, just focusing on work on tetrapods in the Karoo Basin and their geological and paleoenvironmental context, I strongly recommend citing papers like Smith and Botha-Brink (2014), Viglietti et al. (2018), Botha et al. (2020), Viglietti et al. (2021); Gastaldo et al. (2015; 2018, 2019a, 2019b, 2020, 2021). There’s a number of other papers that you could cite dealing with other aspects of the Karoo faunal turnover as well, covering things like changes in bone histology, community structure, etc. Such citations would give a better idea of current thinking on the severity and duration of the extinction, as well as current debates about whether the marine and terrestrial events were synchronous and the degree to which the apparent extinction might be the result of other factors such as lateral facies changes.

Line 270: It would be good to include some citations that lead to the hypothesis that the clades went extinct in this order.

Line 282: insert comma between speciation and extinction

Line 293: change synapside to synapsids

Line 300: This also might be complicated by the geographic shifts in best-sampled areas going from the early to middle Permian (e.g., compare results in Benson and Upchurch 2013 to those in Brocklehurst et al. 2017).

Line 322: Maybe also cite (and use) Schneider et al. (2020), as that is a recent attempt at making global correlations and age estimates for Permo-Carboniferous terrestrial strata.

Figure 2 caption: change to fossil find

Line 342: maybe replace w.r.t. with ‘with regard to’ spelled out

Line 415: I assume here that you are using fossil ages instead of stratigraphic heights to calculate the Strauss and Sadler-style confidence intervals. You should note that in this section to avoid confusion.

Line 418: change since to because

Line 432: This is an interesting observation. Off the top of my head, I’m not aware of studies that have tried to compare the rates of speciation and fossilization in empirical datasets. That might be my ignorance, though. If there are studies that do this, it would be interesting to note how the rates they report compare to what you observed here; i.e., do they have fossilization (or fossil recovery) rates higher than speciation rates?

Line440: change since to because

Table 3 caption: I assume the units of the values presented here are in millions of years. Is that correct? If so, please note that in the caption.

Line 449: change to: “...which we call the mean...”

Figure 3 caption: please specify what the brown ovals represent on branches of the tree. I think these are observed stratigraphic ranges, but it’s not entirely clear whether that’s the case given that some have gaps, and others seem to be made of several overlapping parts.

Line 482: change while to although
Figure 4: The colored sections under the curves are hard to see without really zooming in. I might recommend making this figure into for separate figures (i.e., one for ophiacodontids, one for edaphosaurids, one for sphenacodontids, one for ‘other’ taxa).

Figure 5 caption: change to: “...and Sphenacodontidae base on our dataset.”

Line 498: I think it would be good to note here that the Strauss and Sadler approach (originally, at least) had somewhat different goals: determining confidence intervals on local stratigraphic ranges based on stratigraphic occurrence data in individual sections. So yes, your method works better for your focal question (i.e., confidence intervals on global stratigraphic ranges, taking possible unseen diversification into account), but it is also designed more specifically for that question than the original approach.

Line 516: particularly given the fact that most middle-late Permian tetrapod localities are in higher latitude areas, away from what seems to be the main geographic center of pelycosaur diversity

Lines 522-530: The initial report of dinocephalians in the lowermost Poortjie Member has recently been incorporated into the official revised version of the Tapiniocephalus Assemblage Zone (see Day and Rubidge 2020). Also note that although dinocephalian taxa play a role in defining the lower boundaries of the two Tapinocephalus AZ subzones, the upper boundary of the upper subzone is defined by the first appearance of Endothiodon. Therefore, the absence of dinocephalians is not strictly part of the AZ’s definition.

Line 534: The discussions of the Carboniferous origin of therapsids in Abdala et al. (2008), Spindler et al. (2015), and Angielczyk and Kammerer (2018) would also be good to cite here.

Line 537: Change to: “...debated (Amson and Laurin, 2011; Spindler, 2020) and beyond the scope of this study.” It would also be good to note and discuss Spindler’s (2014) re-identification of potential therapsid material from the Pennsylvanian.

Line 539: I recommend including some citations for the Roadian first appearance of therapsids.

Line 548: I think it would be good to mention that an apparently slow initial radiation of therapsids stands in contrast to Kemp’s hypothesis of a very rapid therapsid diversification (e.g., Kemp 2006, 2009).

Line 554: change to: “...remained rare at that time.”

Lines 565-569: This paragraph and Figures 6-9 don’t really seem to be fully incorporated into the paper. If you plan to include them in the main text, I think you will need to add some discussion of them in the results and discussion sections of the paper. Alternatively, you could have these figures and the current appendices be supplemental data. The latter option makes more sense to me because I think the current structure of the paper is streamlined and logical, whereas adding in the additional content would make it more complex and harder to follow.

---

## Round 0.2 · Minor Revisions

· Academic Editor

Minor Revisions

Thank you for providing such a detailed response to reviewers. I have now received back the comments of two out of the three reviewers from the first round. Both of them were very positive about the manuscript. However, R2 made some final recommendations that I think are worth implementing.

I also would like to make my own comments, briefly: I think the paper could be shortened without losing the main message. Overall, I think the text is a bit verbose. For example, the space devoted to discuss the three problems with the taxic approach in the introduction could be shortened. A few sentences touch upon topics, such as disparity, which are not the goal of the new FBD model you're proposing. The same can be said about the description of the four mass extinctions in the Permian, it could be shortened by e.g., not quoting authors directly or simply making more straightforward statements that help the reader to understand your point, which is basically using an FBD model to estimate extinction and speciation dates and then applying it to a case study. Make an effort to shorten the description of alternative methods to date extinction times: branch-by-branch, global method, previous FBDs models. The results from the two analyses you did and that are in the supplementary material don't need to be fully explained in the text, just mention them in the results, pointing the reader to the suppl mat. Also, try to avoid long citation strings. I know that in some instances they're necessary, but I'm trying to make your paper more concise and as short as possible. It takes a lot of space for you to finally present your new model and test it against real data.

I believe that after these final amendments your paper will be ready to be published.

·

Basic reporting

The text is clear and unambiguous - complex, but professional and thorough. The literature is very well covered (although they might like to add Wang et al. (2016) (Wang, S. C., Everson, P. J., Zhou, H. J., Park, D., and Chudzicki, D. J. (2016). Adaptive credible intervals on stratigraphic ranges when recovery potential is unknown. Paleobiology 42, 240–25) to their discussion that begins on line 101 – I think it is among the best of the simple Strauss and Sadler-like methods. The article is well structured and self contained.

Experimental design

The paper represents original research, well-defined, relevant, and meaningful, and its relationship to previous research is particularly well articulated. The paper is also fully rigorous, performed to an unusually high technical standard. The methods are well described, with, if anything, too much detail (but what constitutes 'too much' is subjective, and it is better to err on the side of too much rather than too little).

Validity of the findings

The findings appear valid. Simple as that.

Additional comments

The authors have done an outstanding job in dealing with my rather copious comments on an older version of the paper. The only thing I would say is that I still found the section on the 4 extinction hypotheses hard going. Maybe right at the outset, just tell us that you are going to start from the oldest to youngest, to give the reader some sort of scaffold to help hold the copious detail, most of which will not be of interest to most readers (unless they are unlucky enough to only have Permian synapsid workers read the paper!).

·

Basic reporting

General comments: This is the second time I’ve reviewed this manuscript, and I appreciate the very detailed response to reviewers that the authors provided. They were clearly very conscientious in addressing the issues that were raised in the previous reviews. I feel they have satisfactorily addressed my previous comments. I included a few minor comments and corrections below, but I think the manuscript is basically ready to accept now.

Title: maybe remove ‘some’

Line 33: Kind of a semantic comment, but regularly implies to me an aspect of periodicity, and I’m not sure that’s what you mean to imply here. Maybe it is, given things like the discussion over the years of things like periodicity in mass extinctions, but if it’s not I think ‘frequently’ would be a better choice.

Line 112: change various sizes to differing richnesses; this will prevent readers from mistakenly thinking that you’re talking about scale instead of diversity

Line 114: change to approaches

Figure 1 caption: change green to yellow

Line 203: following reviewer 1’s recommendation to include the code for the analysis in the supplement, you should call out the supplement here (in addition to github).

Line 288: I appreciate you adding the new information about the Viglietti et al. paper. However, it makes the paragraph seem somewhat confusing because you bounce back and forth between talking about the terrestrial extinction being relatively long and relatively short. Maybe start with the ~10 Mya hypothesis (i.e., Erwin, 1990), and then go from there.

Line 503: I think you mean 290.1

line 506: replace is with are

line 507: replace was with were

line 521: may change to ‘...a single relevant taxon...’

Ling 558: change to compute

Line 560: is their a discussion of the full set of methods you used in the supplement? If so, please call that out here (likewise I would recommend adding it if it’s not currently included).

Line 579: change to require

Line 651: I’m not sure what you mean by ‘quite correct’ here. Maybe something like ‘slightly worse’ would be more accurate.

Line 676: I recommend explicitly restating the hypothesis here to remind readers of what it is.

Line 765: capitalize Permo-Carboniferous

Line 810: change to assessing

Experimental design

see above

Validity of the findings

see above

Additional comments

see above

---

## Round 0.3 · accepted · Accept

· Academic Editor

Accept

Thanks for making these final amendments to the text. I’m pleased to accept your manuscript as is and I anticipate it will have a big impact on the field.